# Existing terminology related to antimicrobial resistance fails to evoke risk perceptions and be remembered

Eva M. Krockow [1✉], Kate O. Cheng[2], John Maltby [1] & Eoin McElroy[3]

## Abstract

**Background** Antimicrobial resistance (AMR) is a global healthcare threat promoted by all use of antibiotics. Hence, reducing overuse of antibiotics is essential. The necessary behaviour change relies on effective public health communication, but previous information campaigns—while showing some successes—have fallen short in generating a lasting increase of public awareness. A potential reason for this is AMR-related terminology, which has been criticised as inconsistent, abstract and difficult to pronounce. We report the first empirical test of word memorability and risk association for the most frequent AMR-related health terms.

**Methods** Across two surveys sampling 237 US and 924 UK participants, we test people's memory for and the risk they associate with six AMR-related terms and thirty-four additional health risk terms (e.g., cancer). Participants also rate the terms on different linguistic dimensions including concreteness, familiarity, processing fluency and pronounceability.

**Results** Our findings suggest that existing AMR-related health terms—particularly "AMR" and "Antimicrobial resistance"—are unsuitable for public health communication, because they score consistently low on both memorability and risk association. Out of the AMR terms, "Antibiotic resistance" and—to a lesser extent—"Drug-resistant infections" perform best. Regression analyses suggest that linguistic attributes (e.g., familiarity, processing fluency, pronounceability) are predictors of the terms' risk association.

**Conclusions** Our findings highlight an urgent need to rename AMR with a memorable term that effectively signals the existential threat of AMR and thereby motivates a change in antibiotic use. The success of the revised term is likely to depend, at least partially, on its linguistic attributes.

### Plain language summary

Antimicrobial resistance (AMR) is a serious public health risk. It means that microorganisms, such as bacteria, change in a way that existing medicines, such as antibiotics, no longer kill them. As a result, it may be impossible to treat even common infections. Increasing the public's understanding of AMR could help avoid its development, but to date, awareness campaigns have not been very successful in changing behaviour. Here, we aimed to understand why, by investigating the language used to communicate about AMR. Participants rated how much health risk they associated with different words (i.e., cancer, Ebola, AMR). People generally found it difficult to remember words associated with AMR and did not think they sounded risky compared to other health risk words. Future risk communication might benefit from renaming AMR to better signal the severity of the problem and motivate behaviour change.

---

[1] School of Psychology and Vision Sciences, University of Leicester, Leicester, UK. [2] Institute of Epidemiology and Health Care, University College London, London, UK. [3] School of Psychology, Ulster University, Coleraine, UK. ✉email: emk12@le.ac.uk

Antimicrobial resistance (AMR) is a global healthcare threat of unprecedented dimensions, which was associated with an estimated 4.95 million deaths in 2019[1], thus vastly surpassing the 3.3 million death toll attributable to COVID-19 in 2020[2]. AMR refers to the biological process whereby microorganisms such as bacteria, viruses and fungi mutate, gaining the ability to resist treatment with medication such as antibiotics. While the mutation of microorganisms is a natural process, every use of antibiotics promotes its occurrence and accelerates the problem. The importance of human choices regarding antibiotic use in human health, animal health and agriculture adds a behavioural challenge of effective resource management to the biomedical problem of AMR[3,4]. Overuse of antibiotics is common and driven by a complex interplay of motivations, including risk aversion and immediate patient concerns in human health[5] and growth promotion in animal health and agriculture[6]. To protect modern medicine and conserve the efficacy of existing medication for future generations, we must address this behavioural challenge and reduce antibiotic use internationally.

A key barrier to tackling AMR is the persistent lack of knowledge and awareness about the issue, especially amongst lay people without more specialised medical knowledge[7–12]. Past media coverage of AMR fell short of communicating the problem's urgency and severity. Reasons for this included competing health risks such as sepsis, which unduly dominated the media due to more easily comprehensible disease narratives[12–14]. Attention to AMR further fell away during the recent COVID-19 pandemic (e.g.,[15]). This is despite scholars labelling AMR as a "silent pandemic"[16], and suggesting that the spread of multi-drug-resistant bacteria and untreatable infections could lead to the next international pandemic[17].

**Previous AMR risk communication**. Increasing stakeholder recognition for the need of more wide-spread AMR risk communication has paved the way for extended information campaigns (e.g.,[18]). Still, existing efforts have been criticised for inadequate use of language and framing[19]. Most importantly, perhaps, the scientific name "antimicrobial resistance" appears to be unsuitable for public health communication (e.g.,[19–21]). It is a term that is inconsistently used (i.e., many alternative terms exist), difficult to pronounce, abstract and has little intuitive meaning (i.e., in contrast with terms like "heart disease", "antimicrobial resistance" evokes few immediate associations of what it might involve). A relevant report by the Wellcome Trust includes a theoretical discussion and comparison of "antimicrobial resistance" and the five most commonly used variants including "AMR", "antibiotic resistance", "bacterial resistance", "drug-resistant infections" and "superbugs".

Across all UK and US media coverage on the topic of AMR, "antibiotic resistance" is the term most commonly used[19]. This is followed by "superbugs", a more colloquial term highlighting the specific role of particularly resistant bacteria, and "antimicrobial resistance", the umbrella term, which combines antibiotic resistance, antiviral resistance and antifungal resistance. The terms "drug-resistant infections" and "bacterial resistance" are also used, but to a lesser extent. Finally, the acronym "AMR", typically used to abbreviate "Antimicrobial Resistance" in the academic literature, is used least frequently out of the six key terms in the UK and US media. An overview of key information about each term is provided in Table 1.

Scholars have called for the consistent use of only one term and suggested "drug-resistant infections" as the most promising one, because it highlights the role of infections[19]. The term "infection" is likely to be meaningful to lay populations[21], and the associated threat could provide a cue to action.

Despite academic support for the term "Drug-resistant infections", no research has conducted a comprehensive empirical test of its effectiveness or compared it to the other existing AMR-related terms. Some qualitative results have highlighted the overuse of technical jargon in the context of AMR communication and indicated lay people's difficulties in making sense of terminology relating to microbes[19], but there is an urgent need for large-scale, quantitative research on the effectiveness of AMR terminology. We aim to fill this gap by reporting results from two related studies pertaining to US and UK lay people's perceptions of the six most commonly used AMR terms, and comparing their overall effectiveness to other health- and disease-related terminology.

**Measuring the effectiveness of terminology**. To support our planned comparison of different health terms' effectiveness, this paragraph will discuss and recommend approaches for measuring effectiveness, and identify theory-based correlates and predictors. Word effectiveness is inherently subjective and difficult to define, but literature from the fields of cognitive psychology and language processing proposed the closely related concept of "word attensity"[22]. This refers to a word's "potential to engage people, capture their attention and improve subsequent memory of the word" ([20], p.2). Memorability therefore appears to be a key factor for effective terminology. Some previous research attempted to derive a lexical measure of memorability, for example calculating the ratio of meaningful characters within brand names[23]. However, with construct validity being difficult to verify, a more direct measure involves empirical assessment of memory performance for the terms in question. Amongst the most reliable word memory measures is Green, Allen, and Astner's[24] Word Memory Test[25], which includes a word recognition test using a multiple choice question format. The current study will employ a similar word recognition test, requiring participants to consider a list of different health risk terms and to indicate whether or not they had previously been presented with a specific term as part of the main study.

Word memorability appears crucial to general word attensity, but a term's effectiveness in the context of health risk communication is likely to be more complex. Models of health behaviour, such as Rosenstock's[26] seminal Health Belief Model, highlights the important role of risk perception, which entails beliefs about both a risk's severity and personal susceptibility or vulnerability to it, in motivating certain health-related behaviours. This theoretical framework suggests that effective disease terminology must be able to evoke risk perceptions and alarm the general population about an impending threat. Risk perception is a subjective concept commonly assessed using Likert rating scales[27]. A few studies have investigated risk perceptions evoked by terminology. For example, Song and Schwarz[28] investigated how the names of food additives and amusement park rides influenced people's perceptions of risk. The study in question required participants to complete 7-point Likert-ratings ranging from "very safe" to "very harmful" or "very risky" for food additives and amusement park rides respectively, and a similar scale was subsequently adopted for the surveys reported in this paper.

In addition to these measures of word effectiveness, previous research has evidenced a number of closely related linguistic dimensions that are likely to be associated with or predict risk association and memorability. These include phonetic aspects such as the pronounceability of a word[29] and "processing fluency", which refers to the general ease experienced when making sense of a word[30]. Additionally, more semantic aspects of terminology may play a role, most notably the related concepts of

**Table 1 Overview (including context and frequency of use) of the six most common AMR-related health terms.**

| Term | Context and frequency of use |
|------|------------------------------|
| AMR | • Acronym for "antimicrobial resistance", most commonly used by experts and communicators and less familiar amongst the general public[19] |
| | • Least frequently used by general media[14] |
| Antibiotic resistance | • Most frequently used term across US media coverage about AMR |
| | • Second most frequently used term across UK media coverage about AMR[14] |
| Antimicrobial resistance | • Umbrella term, which combines antibiotic resistance, antiviral resistance and antifungal resistance |
| | • The term "antimicrobial" previously described as technical or complicated by the public[19] |
| Bacterial resistance | • Variant of "antibiotic resistance", which highlights the role of bacteria |
| Drug-resistant infections | • Highlighted by experts as term with the most potential to communicate existential risk (due to emphasis on infections)[19,21] |
| Superbugs | • Often used in informal and colloquial contexts |
| | • Most frequently used term across UK media coverage about AMR[14] |
| | • Second most frequently used term across US media coverage about AMR |

familiarity and concreteness. Indeed, research suggests that concrete terminology appears familiar to people, evokes clear mental associations and is thus more easily processed and remembered[31]. Again, these linguistic dimensions are difficult to measure. Attempts at objective measures calculated the word length, number of syllables[23] or the relative frequency of a word's phonemes in the base language[32], but these often failed to fully represent the complex variability of language. Consequently, many researchers favour Likert-style self-report measures (e.g.,[33]) to obtain subjective ratings for linguistic dimensions, and we decided to adopt a similar approach in the present research.

**Study aims and research questions**. This study aims to pave the way for an overdue language change in AMR risk communication. Rather than replacing a single AMR-related term, we seek to change the current use of inconsistent language by testing whether any of the existing terminology has the potential to serve as uniform, key term. We report results from two survey studies, which assess the relative effectiveness of existing terms pertaining to the health threat of AMR. Specifically, we test US (Study 1) and UK (Study 2) lay people's risk association and memory for the six most common English language terms used in the context of AMR: "AMR", "Antibiotic resistance", "Antimicrobial resistance", "Bacterial resistance", "Drug-resistant infections" and "Superbugs"[19]. The six terms are further compared with other key health risks and disease names as identified by the WHO's Global Health Risks report[34], examples of which include "cancer", "diabetes", and "high blood glucose".

Finally, we conduct regression analyses of variables that may predict risk association and memorability of AMR-related health terminology. We test models containing the four linguistic variables that have been suggested to correlate with effectiveness (concreteness, familiarity, processing fluency, and pronounceability), while controlling for a range of demographic factors (e.g., age, sex, education level), medical history (e.g., number of recent doctor visits; antibiotic medication history), and vocabulary.

Our research includes two studies: Study 1 reports results from a US sample obtained in 2020 and Study 2 reports results from a UK sample obtained in 2021. The follow-up study was conducted to improve the generalisability of our initial results across different national contexts and at different data collection times. Additionally, Study 2 addressed specific shortcomings of Study 1's sampling approach by using nationally representative sampling techniques and increasing the sample size.

Overall, the two studies address the following research questions:

1. How effective are AMR-related terms in evoking risk associations, and how memorable are they, compared with the names of other key health risks and diseases?

2. What factors predict risk association and memory for AMR-related health terms?

## Methods

**Participants**. The University of Leicester School of Psychology and Vision Sciences Ethics Committee approved both studies reported in this article. All ethical regulations were followed and informed consent obtained from all human participants. For Study 1, US participants were recruited via opportunity sampling through the online platform Amazon Mechanical Turk (MTurk) in May 2020. A total of 305 participants took part in the study, and each received a remuneration of $3.60 ($10.80 pro-rata). Amazon MTurk was previously highlighted as an online data collection platform with a large participant pool, good diversity levels and convenient access to high-quality data[35]. Recent years (2018−2019) have seen rising concerns about increasing numbers of inattentive participants and so-called "bots"[36], and a sudden change during the start of the COVID-19 pandemic, when lockdowns and changes to the job market meant that many people turned to online alternatives to earn an income. While this change resulted in more people joining the MTurk participant pool and further increased sample representativeness, concerns regarding participant attentiveness persisted[37]. Recognising the potential of poor-quality data obtained through Amazon MTurk, we conducted stringent data quality screening (reported below) and implemented a follow-up survey recruited through Prolific, which has repeatedly been shown to produce higher quality data[38]. For Study 2, UK participants were recruited as a nationally representative sample (using the UK Office of National Statistics' quota of age, sex and ethnicity) through the online platform Prolific in November 2021. 998 participants took part in the study and each received a remuneration of £1.88 (£7.52 pro-rata). 68 and 74 participants were excluded from Study 1 and Study 2 respectively, because their responses failed to meet the quality screening criteria (see Fig. 1). Table 2 presents descriptive statistics for participant variables of both studies.

**Design**. Both studies assessed two dependent variables as measures of word effectiveness that were compared across 40 different health terms including the 6 AMR-related terms: risk association and memorability. Risk association was an ordinal score, ranging from 1–7. Memorability was a dichotomous score (1 = correctly remembered; 0 = not remembered).

To test for predictors of the two-word effectiveness measures, four theoretically relevant linguistic measures (familiarity, processing fluency, concreteness, and pronounceability) were assessed in fixed order for each word. Their scores were ordinal data, ranging from 1–7.

We also measured nine control variables. These included six frequently used demographic variables, one of which was continuous (age), while the others were coded as dichotomous variables for ease of analysis: sex (male = 1, female = 0), first language (English = 1, other = 0), race (white = 1, other = 0), education (university educated = 1, non-university educated = 0), work industry (healthcare = 1, other = 0). Two variables pertained to the participants' medical history, which had a potential bearing on antibiotic knowledge. "Doctors visits" contained the levels "yes = 1", confirming that the participant had visited a doctor at least once over the past year, and "no/unknown = 0", indicating the opposite. The variable "antibiotics taken" contained the levels "yes = 1", indicating that a participant had taken antibiotics over the past year, and "no/unknown = 0" indicating the opposite. Finally, a Wechsler Adult Intelligence Scale (WAIS) vocabulary score was calculated for each participant, because previous research showed that cognitive abilities such as reading

skills affected word memory[39]. The score consisted of interval data ranging from 0–50.

**Materials.** All materials were presented via an online questionnaire, and a full set is provided in Supplementary Note 1. The questionnaires were almost identical for Study 1 and Study 2, with minor adaptations being made to account for differences in US and UK education systems and racial composition. Below is a brief description of each section of the questionnaire. The first section assessed demographic information and a brief medical history of participants. The main section presented participants sequentially with 40 target words and 2 filler words. In Study 1, the terms were presented in one of ten orders, created using the rand() function in *Excel*. In Study 2, the order of presentation was determined by the pseudorandom shuffle function of *Gorilla*, using a different "seed" for each participant. The 40 health terms included the six most common English language terms used in the context of AMR, i.e., "AMR", "Antibiotic resistance", "Antimicrobial resistance", "Bacterial resistance", "Drug-resistant infections" and "Superbugs"[19]. 32 additional terms were chosen from the WHO's Global Health Risks report[34] and an online spotlight article highlighting ten threats to global health in 2019[40]. Finally, we added "COVID-19" and "coronavirus" due to topicality at the time. The total list of terms represents some of the biggest health threats faced by global society at the time the survey was conducted, and associated mortalities with each term are presented in Supplementary Data 1. Out of the 40 health-related terms, 25 consisted of singular words (e.g., "diabetes", "superbugs" and "drought"), ten consisted of multiple words (e.g., "high blood glucose", "heart disease" and "bacterial resistance") and five represented acronyms or abbreviations (e.g., "COVID-19", "AIDS" and "AMR"). For attention checking, the words "table" and "school" were included as non-health-related filler words of high frequency, familiarity and easy pronounceability.

Participants rated all 40 health terms and the two filler items on the dimension of risk association and the four linguistic dimensions using a 7-point Likert scale (see Supplementary Note 1 for detailed instructions and full definitions of the measures). The third section of the questionnaire consisted of a

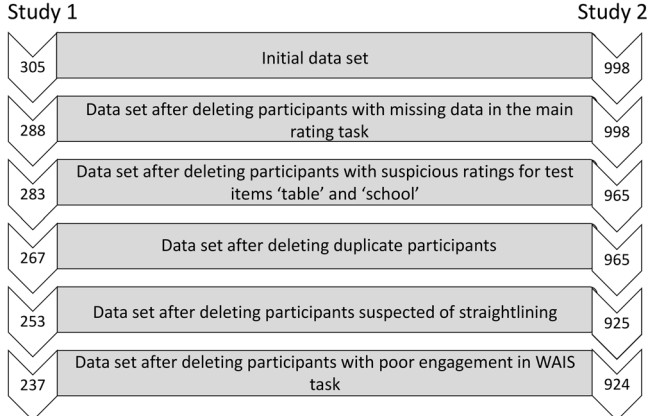

**Fig. 1 Process of data screening for Study 1 and Study 2.** Process of data screening: The different levels display the numbers of remaining participants for Study 1 and Study 2 after applying each screening criterion. More details on the screening process are provided in the Statistics and Reproducibility section.

| Table 2 Descriptive statistics for participant variables across both studies. | | | |
|---|---|---|---|
| | | **Study 1** **n = 237, US opportunity sample via MTurk** | **Study 2** **n = 924, UK nationally representative sample via Prolific** |
| **Age (in years)** | | **M = 36.62; SD = 10.72** | **M = 45.00; SD = 15.80** |
| **Vocabulary score (maximum of 50)** | | **M = 32.68; SD = 11.09** | **M = 34.59; SD = 7.76** |
| Sex: | Female | 39.2% | 47.7% |
| | Male | 60.8% | 51.8% |
| | Other/not specified | 0% | 0.4% |
| Race | White | 67.5% | 85.1% |
| | Other | 32.5% | 14.9% |
| Education | University educated | 80.6% | 56.4% |
| | Non-University educated | 19.4% | 43.6% |
| First language | English | 96.2% | 88.9% |
| | Other | 3.8% | 11.1% |
| Work Industry | Healthcare | 8.9% | 13.2% |
| | Other | 91.1% | 86.8% |
| Doctors' visit in the past 12 months? | Yes | 66.7% | 57.1% |
| | No/unknown | 33.3% | 42.9% |
| Antibiotics taken in the past 12 months? | Yes | 43.5% | 22.7% |
| | No/unknown | 56.5% | 77.3% |

Means (M) and standard deviations (SD) are displayed for continuous variables and percentages are displayed for categorical variables. For ease of analysis, categorical variables such as race, education, first language and healthcare were dichotomised (see "Design" section).

verbal comprehension index, namely the vocabulary subtest of the Wechsler Adult Intelligence Scale-IV (WAIS-IV). This test assessed people's vocabulary as a type of crystallised intelligence by asking them to define progressively more difficult words taken from the English language. In the context of the current study, the test was also used as a filler task before the subsequent memory test.

Finally, the surveys concluded with a memory test. Participants were presented with a list of 80 items containing the previous 40 health terms and 40 health-related distractor items (see Supplementary Note 1). For each item on the list, participants had to indicate whether or not they remembered having been presented with the respective word during the main task.

**Procedure**. Participants were informed the study purpose was to examine how different people think about different health risks. They signed an electronic consent form and subsequently completed the questionnaire in their own time. On average, the questionnaire took between 10–20 minutes to complete. In Study 2, participants were timed out if they took longer than one hour to complete the questionnaire. All participants were reimbursed with $3.60 and £1.88 respectively for Study 1 and Study 2 within three days of participation.

**Statistics and reproducibility**. All data were rigorously vetted prior to the analyses. This included an attention check of participants by screening responses to the test filler items "school" and "table". Participants with suspicious scores (i.e., those who gave these deliberately simple items low ratings of 1 or 2 on familiarity and pronounceability) were removed. Furthermore, responses to the WAIS vocabulary task were used to detect duplicate participants. The task required free word input from participants, and different participant submissions with identical responses across all items were discarded as duplicates (see Fig. 1). To calculate numerical WAIS scores from the free-text data, the official WAIS scoring sheet was used and responses were manually assigned ordinal scores ranging between 0 and 2, with 0 indicating a fail and 2 indicating a perfect response. The scores across all items were summed for each participant to calculate the overall score. For Study 1, it became apparent that some text had been copied and pasted from online sources such as Wikipedia. Obvious examples of such literal duplication were scored as "0". When examining the distribution of WAIS scores, we noted a comparatively high number of very low scores—particularly in the US sample of Study 1. This caused concern, because even individuals with an elementary grasp of the English language should be able to provide definitions for the first few test items (i.e., "apple", "glove", "breakfast" and "curious"). We therefore decided that a very low WAIS score of <5 was an indicator of either (1) very poor English language skills or (2) very poor study engagement, both of which we considered additional grounds for exclusion from the study. Finally, all Likert ratings for the 40 health terms were screened for straightlining. For each term, the standard deviation of Likert scores was calculated across the items concreteness, familiarity, fluency, pronounceability and risk perception. Subsequently, we calculated the mean standard deviation across all terms. Individuals with scores of ≤.04 were excluded on suspicion of straightlining.

All analyses reported were two-tailed. We first checked the assumptions for parametric tests to compare the scores for risk association across the six AMR-related health terms. The assumption of sphericity was violated throughout, which is why we chose Greenhouse–Geisser adjustments for all ANOVAs. To compare the binary memory scores across the six AMR-related health terms, Cochran's $Q$ tests were used. We conducted two-tailed Bivariate Pearson's correlations for all continuous variables and point-biserial correlations for variable pairings containing a dichotomous variable to test for significant correlations between all dependent and independent variables (both main predictors and controls). Assumptions of linearity, no significant outliers and normality for Pearson's correlations were checked through inspection of scatterplots. The assumption of equal variances for point-biserial correlations was tested using Levene's test of equality of variances. We conducted separate regression analyses to test for statistically significant predictors of word effectiveness measures (risk association and memorability) of AMR-related health terms. All regressions were hierarchical (i.e., predictors included in theoretically determined blocks). The first block consisted of demographic control variables and vocabulary score, the word effectiveness measure was entered in block 2, and linguistic variables were introduced in block 3. For the continuous DV of risk association, we ran multiple linear regressions and checked assumptions of (1) multivariate normality by plotting the residuals, (2) multicollinearity by checking the VIF scores and homoscedasticity by plotting the standardised residuals against their predicted values. We ran binomial logistic regressions for the binary variable of memorability. The reference groups of dichotomous variables used in the regression analyses were those coded as "1" (see "Design" section).

## Results

**Comparison of risk association and memorability scores**. An overview of the risk association ratings and memory scores for the 40 health risk and disease terms across Study 1 and Study 2 is provided in Figs. 2 and 3.

The six AMR-related health terms scored consistently low compared to the other health risk and disease terms. With regard to risk association, terms such as "cancer", "Ebola" and "heart disease" were rated as the riskiest with mean scores of up to 6.65 (out of 7) for "cancer" in Study 2, while "chicken pox", "AMR" and "diarrhoea" were rated as having the lowest risks, with a mean score as low as 4.13 for "AMR" in Study 2. Three of the six AMR-related health terms ("Bacterial resistance", "antimicrobial resistance" and "AMR") fell within the lowest-scoring quintile of risk association.

A Greenhouse–Geisser-corrected within-subjects ANOVA found significant differences between the risk association ratings for the six AMR-related health terms in both Study 1, $F(4.647, 1096.740) = 38.302$, $p = 4.0731^{E-34}$, $\eta_p^2 = 0.140$, and Study 2, $F(4.762, 4395.380) = 407.222$, $p = 0.0^{E0}$, $\eta_p^2 = 0.306$. Mean ratings are displayed in Table 3. Pairwise comparisons using Bonferroni adjustment for multiple comparisons are reported in Supplementary Tables 1 and 2. Across both studies, ratings for "AMR" were significantly lower than all other terms. Ratings for "Antimicrobial resistance" were significantly lower than all other terms apart from "AMR" and "Bacterial resistance" (only in Study 1). Finally, "Drug-resistant infections" had significantly higher risk association ratings than all other AMR-related health terms across both studies.

With regard to memorability, "Diarrhoea", "HIV" and "AIDS" were all highly memorable, with more than 96% of Study 2 participants correctly recalling these terms. Again, AMR-related health terms scored comparatively low; "AMR", "Antimicrobial resistance" and "Drug-resistant infections" scored within the lowest quintile, and "Antibiotic resistance" was the only term to score in the top half (ranking 23rd and 18th most memorable in Study 1 and Study 2 respectively).

A Cochran's $Q$ test was conducted to test for significant differences in the proportions of participants who correctly remembered the six AMR-related health terms (see Table 3). A

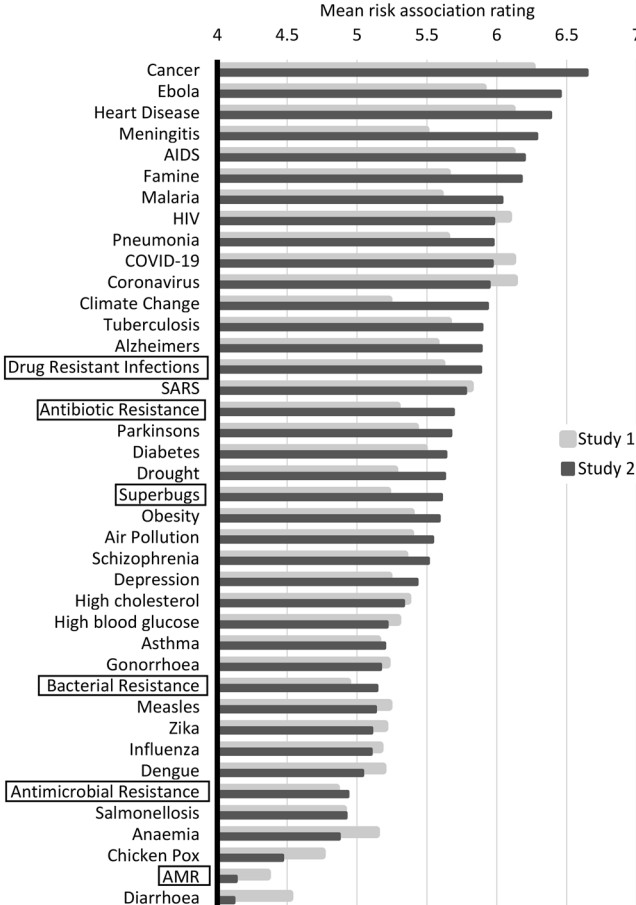

**Fig. 2 Mean risk association ratings for all 40 health risk and disease terms in Study 1 and Study 2.** Mean risk association ratings for all 40 health risk and disease terms. The horizontal axis shows mean ratings on a 7-point Likert scale, with smaller scores indicating lower risk ratings. The vertical axis shows the 40 health risk and disease terms, ordered highest to lowest on risk association based on the results from Study 2. AMR-related health terms are highlighted with black frames. Results from Study 1 ($n = 237$) are depicted using grey bars and results from Study 2 ($n = 924$) using black bars. Source data for the figure are provided in the Supplementary Data 2.

significant difference was found in Study 1, $\chi^2(5) = 25.077$, $p = 0.000135$, and Study 2, $\chi^2(5) = 75.578$, $p = 7.0461\text{E-15}$. Across both studies, the proportions were highest for "Antibiotic resistance", and this was followed by "Bacterial resistance", while "AMR" and "Drug-resistant infections" were lowest. Bonferroni-adjusted pairwise comparisons (see Supplementary Tables 3 and 4) showed that memorability of "Antibiotic resistance" was significantly better than for "AMR" and "Drug-resistant infections" across both studies.

**Predicting word effectiveness of AMR-related health terms.** Following our comparison of word effectiveness for the six AMR-related health terms, we aimed to identify significant predictors of word effectiveness. In our analyses, we focused on two of the AMR-related health terms; "Drug-resistant infections" and "Antibiotic resistance". The former was chosen due to the term's support by previous literature (Mendelson et al.[21]; Wellcome Trust[19]) and because it achieved the highest risk association ratings across both our studies. The latter was chosen, because the term performed best across both word effectiveness measures in Studies 1 and 2.

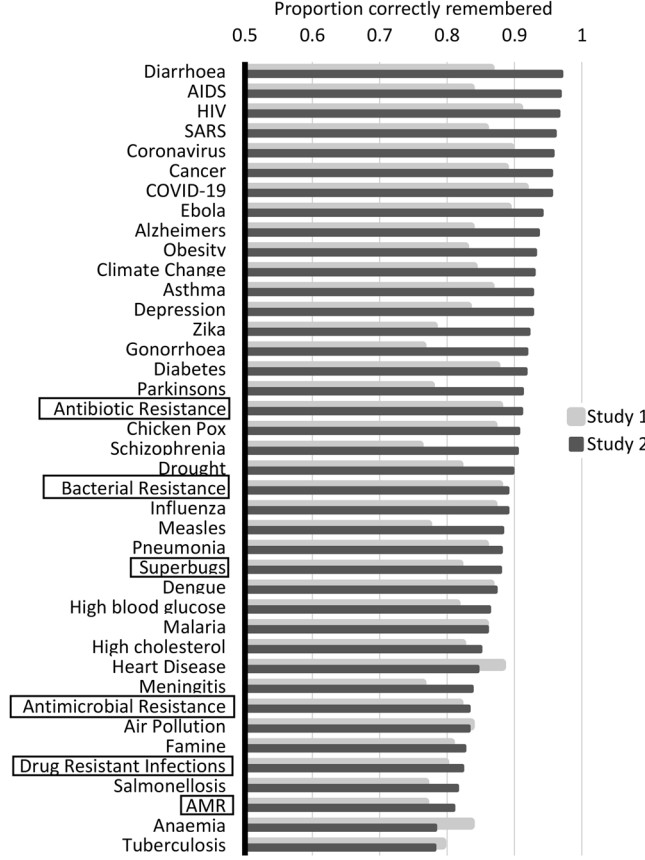

**Fig. 3 Proportion of participants who correctly remembered each of the 40 health risk and disease terms in Study 1 and Study 2.** Proportion of participants who correctly remembered each of the 40 health risk and disease terms. The horizontal axis shows the proportion. The vertical axis shows the 40 health risk and disease terms, ordered highest to lowest based on the results from Study 2. AMR-related health terms are highlighted with black frames. Results from Study 1 ($n = 237$) are depicted using grey bars and results from Study 2 ($n = 924$) using black bars. Source data for the figure are provided in the Supplementary Data 3.

*Regression analyses for the term "Drug-resistant infections".* Prior to our regression analyses presented below, we tested for significant correlations between the two-word effectiveness measures (i.e., risk association and memorability), the different linguistic dimensions (i.e., concreteness, familiarity, processing fluency and pronounceability) and all participant variables including demographic variables, medical history variables and the vocabulary scores. The results are presented in Supplementary Data 4 and 5.

Subsequently, multiple regression analyses were conducted for the dependent variable of risk association, with predictors entered hierarchically in blocks. The first block consisted of nine demographic control variables and explained 13.6% of the variance in risk association in Study 1 and 7.7% in Study 2. Block 2 consisted of the memory score and added non-significant contributions of 0.2% in Study 1 and 0.3% in Study 2. The third block consisted of the four linguistic dimension variables and added 10.2% in Study 1 and 7.4% in Study 2. The full model of 14 predictors was significant in both studies (see Table 4). In Study 1, it explained 24% of the variance in risk association scores ($F (14, 217) = 4.89$, $p = 7.5612\text{E-8}$, adjusted $R^2 = 0.19$), with three individual predictors—vocabulary, familiarity, and pronounceability—making unique contributions, and positively predicting the outcome. In Study 2, our model explained 15.5% of the variance in risk association scores ($F (14, 878) = 11.46$,

$p = 1.5379^{E-24}$, adjusted $R^2 = 0.14$), with four individual predictors—vocabulary, familiarity, processing fluency and pronounceability making unique contributions. Better vocabulary and higher linguistic ratings for familiarity, processing fluency and pronounceability all predicted higher risk association.

Hierarchical binomial logistic regression analyses were conducted with three blocks for memorability. The model was only significant for Study 1. The first block of demographic variables explained 16.5% of the variance in memorability in Study 1.

The second block, which contained the risk association rating, explained an additional 0.5% of variance. The third block of linguistic dimensions added another 3.1%. The full model of 14 predictors (see Table 5) explained 20.1% of the variance in memorability ($\chi^2(14) = 31.031$, $p = .005$, Nagelkerke $R^2 = .201$), with one individual predictor—sex—making a unique contribution. Males were 2.9 times more likely to recall the term "Drug-resistant infections" than females.

*Regression analyses for the term "Antibiotic resistance".* Again, hierarchical multiple regression analyses were conducted for risk association. Neither the first block of demographic variables nor the second block consisting of the memory score made a significant contribution to explaining the risk association score for "Antibiotic resistance" in Study 1. The third block (consisting of the four linguistic dimension variables) accounted for 12.5% of the variance in Study 1. For Study 2, the first block explained 5.8% of the variance, the second block an additional 0.8% and the third block an additional 6.1%. The full model of 14 predictors was significant in both studies (see Table 6). In Study 1, it explained 12.5% of the variance in risk association scores ($F(14, 217) = 2.21$, $p = 0.008$, adjusted $R^2 = 0.07$), with two individual predictors—antibiotics taken last year and familiarity—making unique contributions. Not having taken antibiotics over the past 12 months and a higher linguistic rating of familiarity predicted higher risk association ratings. In Study 2, it explained 13.6% of the variance in risk association scores ($F(14, 878) = 9.90$, $p = 8.1718^{E-21}$, adjusted $R^2 = 0.12$) with four individual predictors—vocabulary, memorability, familiarity and processing fluency—making unique contributions. A better vocabulary score,

---

**Table 3 Descriptive statistics including means (M) and standard deviations (SD) for risk association and proportions for memorablity across the 6 AMR-related health terms in Study 1 and Study 2.**

|  |  | Risk association M (SD) | Memorability *Proportion correctly remembered* |
|---|---|---|---|
| AMR | Study 1 | 4.36 (1.10) | 0.77 |
|  | Study 2 | 4.13 (0.60) | 0.81 |
| Antibiotic resistance | Study 1 | 5.29 (1.33) | 0.88 |
|  | Study 2 | 5.69 (1.22) | 0.91 |
| Antimicrobial resistance | Study 1 | 4.85 (1.32) | 0.82 |
|  | Study 2 | 4.93 (1.25) | 0.83 |
| Bacterial resistance | Study 1 | 4.93 (1.47) | 0.88 |
|  | Study 2 | 5.14 (1.37) | 0.89 |
| Drug-resistant infections | Study 1 | 5.61 (1.25) | 0.80 |
|  | Study 2 | 5.88 (1.20) | 0.82 |
| Superbugs | Study 1 | 5.22 (1.38) | 0.82 |
|  | Study 2 | 5.60 (1.26) | 0.88 |

---

**Table 4 Hierarchical multiple regression analyses with three blocks for risk association for the term "drug-resistant infections".**

|  | Study | b | β | t | p | 95% CI Lower | 95% CI Upper |
|---|---|---|---|---|---|---|---|
| Constant | S1 | 1.59 | -- | 2.10 | 0.04 | 0.10 | 3.08 |
|  | S2 | 2.43 | -- | 6.94 | 0.00 | 1.75 | 3.12 |
| Age | S1 | 0.01 | 0.11 | 1.80 | 0.07 | 0.00 | 0.03 |
|  | S2 | 0.00 | 0.06 | 1.85 | 0.06 | 0.00 | 0.01 |
| Sex | S1 | −0.28 | −0.11 | −1.78 | 0.08 | −0.60 | 0.03 |
|  | S2 | 0.00 | 0.00 | 0.02 | 0.98 | −0.15 | 0.15 |
| Race | S1 | −0.04 | −0.02 | −0.27 | 0.79 | −0.37 | 0.28 |
|  | S2 | 0.15 | 0.04 | 1.32 | 0.19 | −0.07 | 0.36 |
| Education | S1 | −0.03 | −0.01 | −0.14 | 0.89 | −0.42 | 0.37 |
|  | S2 | 0.09 | 0.04 | 1.13 | 0.26 | −0.07 | 0.25 |
| Work industry | S1 | −0.12 | −0.03 | −0.45 | 0.65 | −0.64 | 0.40 |
|  | S2 | −0.03 | −0.01 | −0.23 | 0.81 | −0.24 | 0.19 |
| First language | S1 | 0.43 | 0.07 | 1.06 | 0.29 | −0.37 | 1.23 |
|  | S2 | 0.20 | 0.05 | 1.56 | 0.12 | −0.05 | 0.44 |
| Antibiotics taken last year? | S1 | 0.07 | 0.03 | 0.37 | 0.71 | −0.29 | 0.42 |
|  | S2 | −0.16 | −0.05 | −1.67 | 0.10 | −0.34 | 0.03 |
| Doctor visited last year? | S1 | −0.26 | −0.10 | −1.41 | 0.16 | −0.62 | 0.10 |
|  | S2 | 0.06 | 0.03 | 0.77 | 0.44 | −0.10 | 0.22 |
| Vocabulary | S1 | 0.03 | 0.28 | 4.17 | 0.00 | 0.02 | 0.05 |
|  | S2 | 0.02 | 0.10 | 2.84 | 0.00 | 0.00 | 0.03 |
| Memorability | S1 | 0.13 | 0.04 | 0.66 | 0.51 | −0.27 | 0.53 |
|  | S2 | 0.13 | 0.04 | 1.34 | 0.18 | −0.06 | 0.32 |
| Concreteness | S1 | 0.07 | 0.10 | 1.57 | 0.12 | −0.02 | 0.16 |
|  | S2 | 0.02 | 0.04 | 1.17 | 0.24 | −0.02 | 0.06 |
| Familiarity | S1 | 0.20 | 0.23 | 3.00 | 0.00 | 0.07 | 0.33 |
|  | S2 | 0.09 | 0.13 | 3.55 | 0.00 | 0.04 | 0.15 |
| Processing fluency | S1 | −0.05 | −0.06 | −0.69 | 0.50 | −0.21 | 0.10 |
|  | S2 | 0.13 | 0.13 | 2.88 | 0.00 | 0.04 | 0.21 |
| Pronounceability | S1 | 0.20 | 0.21 | 2.75 | 0.01 | 0.06 | 0.34 |
|  | S2 | 0.12 | 0.08 | 2.09 | 0.04 | 0.01 | 0.24 |

The table shows unstandardised and standardised betas, *t* values, significance levels, and confidence intervals (CI) for standardised beta weights across Study 1 (S1) and Study 2 (S2). Coefficients are from Block 3.

**Table 5 Hierarchical binomial logistic regression with three blocks for memorability for the term "Drug-resistant infections" in Study 1 and Study 2.**

|  | Study | b | p | Exp(B) | 95% CI Lower | 95% CI Upper |
|---|---|---|---|---|---|---|
| Age | S1 | −0.01 | 0.56 | 0.99 | 0.95 | 1.03 |
|  | S2 | 0.00 | 0.93 | 1.00 | 0.99 | 1.01 |
| Sex | S1 | −1.08 | 0.01 | 0.34 | 0.14 | 0.80 |
|  | S2 | 0.06 | 0.74 | 1.06 | 0.74 | 1.51 |
| Race | S1 | 0.11 | 0.80 | 1.11 | 0.50 | 2.50 |
|  | S2 | 0.10 | 0.71 | 1.10 | 0.65 | 1.88 |
| Education | S1 | 0.50 | 0.42 | 1.64 | 0.49 | 5.45 |
|  | S2 | −0.13 | 0.50 | 0.88 | 0.61 | 1.28 |
| Work industry | S1 | 0.26 | 0.66 | 1.30 | 0.41 | 4.12 |
|  | S2 | 0.02 | 0.95 | 1.02 | 0.61 | 1.71 |
| First language | S1 | 0.28 | 0.81 | 1.32 | 0.13 | 13.16 |
|  | S2 | 0.20 | 0.51 | 1.22 | 0.67 | 2.23 |
| Antibiotics taken last year? | S1 | 0.52 | 0.23 | 1.68 | 0.71 | 3.98 |
|  | S2 | 0.20 | 0.34 | 1.23 | 0.80 | 1.88 |
| Doctor visited last year? | S1 | −0.29 | 0.51 | 0.74 | 0.31 | 1.81 |
|  | S2 | −0.12 | 0.53 | 0.89 | 0.61 | 1.29 |
| Vocabulary | S1 | 0.03 | 0.11 | 1.03 | 0.99 | 1.06 |
|  | S2 | 0.01 | 0.64 | 1.01 | 0.98 | 1.03 |
| Risk association | S1 | 0.14 | 0.41 | 1.15 | 0.83 | 1.60 |
|  | S2 | 0.10 | 0.18 | 1.11 | 0.95 | 1.29 |
| Concreteness | S1 | −0.22 | 0.09 | 0.81 | 0.62 | 1.04 |
|  | S2 | −0.04 | 0.39 | 0.96 | 0.87 | 1.05 |
| Familiarity | S1 | −0.04 | 0.82 | 0.96 | 0.70 | 1.33 |
|  | S2 | −0.02 | 0.70 | 0.98 | 0.86 | 1.10 |
| Processing fluency | S1 | 0.03 | 0.87 | 1.03 | 0.71 | 1.50 |
|  | S2 | 0.04 | 0.72 | 1.04 | 0.85 | 1.26 |
| Pronounceability | S1 | 0.21 | 0.21 | 1.23 | 0.89 | 1.70 |
|  | S2 | 0.19 | 0.12 | 1.21 | 0.95 | 1.54 |
| Constant | S1 | 0.32 | 0.82 | 1.38 |  |  |
|  | S2 | −0.54 | 0.49 | 0.58 |  |  |

The table shows betas values, significance levels, odds ratios and confidence intervals (CI) for the adjusted odds ratios. Coefficients are from Block 3.

correct word recall and higher linguistic ratings of familiarity and processing fluency predicted higher risk association.

Again, hierarchical binomial logistic regression analyses were conducted with three blocks for memorability. For Study 1, the predictor "First language" showed complete separation; only nine individuals were non-native English speakers and all of those correctly remembered "Antibiotic resistance". As a consequence, this variable was removed from the model for Study 1. The first block of demographic variables explained 16.2% of the variance in memorability in Study 1 and 4.0% in Study 2. The second block (risk association) explained an additional 1.4% of variance in Study 1 and 1.9% in Study 2. The third block of linguistic dimensions added another 1.7% in Study 1 and 2.4% in Study 2. The full model of 14 predictors (see Table 7) reached statistical significance across both studies. For Study 1, it explained 20.4% of the variance in memorability ($\chi^2(14) = 24.25$, $p = 0.043$, Nagelkerke $R^2 = 0.204$), with one individual predictor—vocabulary—making a unique contribution ($p < 0.05$). Having a higher vocabulary score was a predictor of better memory performance. For Study 2, it explained 8.3% of the variance in memorability ($\chi^2(14) = 33.65$, $p = 0.002$, Nagelkerke $R^2 = 0.083$), with four individual predictors—antibiotics taken last year, doctor visited last year, risk association and familiarity—making unique contributions. People who had taken antibiotics or who had not seen their doctor and those who rated "Antibiotic resistance" as riskier and more familiar were more likely to remember the term.

## Discussion

The two studies reported in this article provide a comprehensive investigation of lay people's perceptions and responses to the health terminology most commonly used to communicate the global threat of AMR. We aimed to test whether a single existing term could replace the use of inconsistent terminology, which is typical of current health risk communication and media reporting. Additionally, we conducted a detailed analysis of correlates and predictors of word effectiveness for the two best-performing AMR-related health terms; "Antibiotic resistance" and "Drug-resistant infections". The effectiveness of health-related terminology was assessed by two measures. These included self-reported risk perceptions associated with the words in question and the terms' memorability (assessed by a prompted recognition task).

Comparing word effectiveness measures of "AMR", "Antibiotic resistance", "Antimicrobial resistance", "Bacterial resistance", "Drug-resistant infections" and "Superbugs" with those of other major health risk and disease terms, the six AMR-related health terms consistently scored low across both studies. Indeed, "AMR" and "Antimicrobial resistance" were amongst the lowest-scoring terms out of the 40 health risk terms for both risk association and memorability, and this was consistent across Study 1 and Study 2. Consulting existing mortality data reported for all 40 health terms (see Supplementary Data 1), which might be considered as a proxy for objective health risk, it becomes evident that the general public's risk perceptions are miscalibrated. While participants correctly judged heart disease and cancer to be among the largest health threats, they severely overestimated the risks of tropical diseases such as Ebola and Malaria, while underestimating the threat of AMR, which ranked 6th in terms of global deaths incurred and is predicted to overtake cancer as a leading cause of death by 2050[41].

Our findings support the previous literature on the inappropriateness of current terminology for public health communication about AMR (e.g.,[19–21]). Statistical comparisons across both studies of the six AMR-related health terms indicated that "Drug-resistant infections" was significantly more effective in inducing risk perceptions than all other existing AMR terms, thus confirming previous theoretical predictions (e.g.,[19]). However, the results of both studies also showed that "Drug-resistant infections" ranked particularly low on memorability, while "Antibiotic resistance" was remembered most easily. These findings suggest that "Drug-resistant infections" surpasses the other existing terms only on one dimension of word effectiveness, while "Antibiotic resistance" appears to be the most effective AMR-related health term overall.

Aiming to increase our understanding of the higher effectiveness of "Drug-resistant infections" and "Antibiotic resistance" and to identify predictors of word effectiveness for enhancing future communications, we conducted a series of regression analyses.

Significant predictor variables of risk association scores varied slightly across the two studies and two AMR-related terms. However, they were consistently predicted by one or more of the linguistic attributes assessed. Higher perceptions of word familiarity but also higher ratings of processing fluency and pronounceability emerged as significant predictors of people's risk perceptions. Additionally, some demographic variables appeared to play a role. A participant's vocabulary range was the most consistent demographic predictor of risk association; participants with better scores rated the risk of AMR-related terms as higher.

These findings confirm previous literature stating that linguistic aspects matter for the effectiveness of terminology in the context of risk communication (e.g.,[28]), while highlighting the importance of additional demographic factors including a person's range of vocabulary. Yet, with our regression models only explaining between 13.6−24% of the variance in risk association scores, future research is needed to identify additional predictors.

**Table 6 Hierarchical multiple regression analyses with three blocks for risk association for the term "Antibiotic resistance".**

|  | Study | b | β | t | p | 95% CI Lower | 95% CI Upper |
|---|---|---|---|---|---|---|---|
| Constant | S1 | 4.58 |  | 5.28 | 0.00 | 2.87 | 6.30 |
|  | S2 | 2.06 |  | 5.14 | 0.00 | 1.27 | 2.84 |
| Age | S1 | −0.01 | −0.05 | −0.73 | 0.47 | −0.02 | 0.01 |
|  | S2 | 0.01 | 0.07 | 1.95 | 0.05 | 0.00 | 0.01 |
| Sex | S1 | −0.19 | −0.07 | −1.05 | 0.30 | −0.54 | 0.17 |
|  | S2 | −0.14 | −0.06 | −1.83 | 0.07 | −0.30 | 0.01 |
| Race | S1 | 0.14 | 0.05 | 0.74 | 0.46 | −0.24 | 0.52 |
|  | S2 | 0.21 | 0.06 | 1.80 | 0.07 | −0.02 | 0.43 |
| Education | S1 | 0.02 | 0.01 | 0.10 | 0.92 | −0.43 | 0.47 |
|  | S2 | 0.12 | 0.05 | 1.44 | 0.15 | −0.04 | 0.28 |
| Work industry | S1 | −0.13 | −0.03 | −0.42 | 0.68 | −0.73 | 0.48 |
|  | S2 | 0.04 | 0.01 | 0.35 | 0.73 | −0.18 | 0.26 |
| First language | S1 | 0.02 | 0.00 | 0.05 | 0.96 | −0.87 | 0.92 |
|  | S2 | 0.16 | 0.04 | 1.22 | 0.22 | −0.10 | 0.42 |
| Antibiotics taken last year? | S1 | −0.46 | −0.17 | −2.29 | 0.02 | −0.86 | −0.06 |
|  | S2 | −0.11 | −0.04 | −1.18 | 0.24 | −0.30 | 0.08 |
| Doctor visited last year? | S1 | −0.07 | −0.02 | −0.33 | 0.74 | −0.48 | 0.34 |
|  | S2 | 0.07 | 0.03 | 0.81 | 0.42 | −0.09 | 0.23 |
| Vocabulary | S1 | 0.01 | 0.07 | 0.99 | 0.32 | −0.01 | 0.03 |
|  | S2 | 0.01 | 0.09 | 2.46 | 0.01 | 0.00 | 0.02 |
| Memorability | S1 | −0.51 | −0.12 | −1.74 | 0.08 | −1.08 | 0.07 |
|  | S2 | 0.28 | 0.06 | 1.99 | 0.05 | 0.00 | 0.55 |
| Concreteness | S1 | −0.05 | −0.07 | −0.97 | 0.33 | −0.15 | 0.05 |
|  | S2 | 0.00 | 0.00 | 0.07 | 0.94 | −0.04 | 0.04 |
| Familiarity | S1 | 0.29 | 0.29 | 3.50 | 0.00 | 0.13 | 0.46 |
|  | S2 | 0.14 | 0.15 | 3.82 | 0.00 | 0.07 | 0.22 |
| Processing fluency | S1 | −0.02 | −0.02 | −0.24 | 0.81 | −0.20 | 0.15 |
|  | S2 | 0.16 | 0.12 | 2.56 | 0.01 | 0.04 | 0.28 |
| Pronounceability | S1 | 0.00 | 0.00 | 0.00 | 1.00 | −0.17 | 0.17 |
|  | S2 | 0.07 | 0.04 | 0.94 | 0.35 | −0.07 | 0.20 |

The table shows unstandardised and standardised betas, t values, significance levels, and confidence intervals (CI) for standardised beta weights across Study 1 (S1) and Study 2 (S2). Coefficients are from Block 3.

The findings for predictors of the memorability score were more complex due to larger variations between studies and different AMR-related terms. While male sex predicted better memory of the term "Drug-resistant infections" in Study 1, no significant regression model could be identified for Study 2. For the term "Antibiotic resistance", memory was predicted by a higher vocabulary score in Study 1 and by a person's recent medical history as well as risk and familiarity ratings in Study 2. Future research is necessary to investigate additional predictors. Furthermore, different types of memory tests could be used. In our study, correct word recognition was high throughout, which may have been due to the fact that false positives were neither assessed nor penalised and participants did not have an incentive (beyond their personal motivation) to respond accurately. Additionally, the word recognition of AMR-related health terms may have been unduly boosted because of several very similar sounding terms (e.g., "Antibiotic resistance", "Antimicrobial resistance" and "Bacterial resistance"). To identify more reliable and consistent predictors of memory performance, future studies could address these methodological points around assessment.

The combined results from our two survey studies in the US and UK suggest that no existing AMR-related health term is likely to be sufficient for drawing the warranted attention to AMR's global public health issue. "AMR" and "Antimicrobial resistance" appear particularly unsuitable for public health risk communication, but it is questionable whether other existing terms can offer effective alternatives. The term "Drug-resistant infections", which was recommended to replace all other AMR-related health terms in future communications[19,21], appears to have some advantages due to its ability to evoke higher risk perceptions, but —notably—these risk perceptions still scored lower than other health risk and disease terms with similar death tolls (e.g., obesity). Additionally, its low memorability questions the term's overall effectiveness. Indeed, if considering both risk association and memorability ratings, the term "Antibiotic resistance" could be considered superior to "Drug-resistant infections". Yet again, "Antibiotic resistance" falls short in its effectiveness compared to other major health risk terms.

Our findings have important implications for health risk communication by a large number of international and national health organisations (e.g., WHO, national governments and public health institutions, global and local charities), media outlets as well as healthcare providers. It is evident that more attention needs to be paid to the specific language used when communicating about AMR. Unified agreements and guidelines towards a consistent use of terminology would be an important step towards achieving this.

Given our findings of the limited effectiveness of all existing AMR-related terminology, future research is urgently needed to identify a different name, which is easy to remember and successful in evoking proportionate risk perceptions. Our research suggests that satisfying certain linguistic criteria (e.g., familiarity, processing fluency and pronounceability) could be a key step in achieving this. In order to secure support from diverse international stakeholders including the health organisations listed above, this research is likely to depend on an iterative process of stakeholder engagement and consensus building. Lessons might be learned from recent success stories including the renaming of the "Wuhan novel coronavirus" into "COVID-19"[42].

**Table 7 Hierarchical binomial logistic regression with three blocks for memorability for the term "Antibiotic resistance" in Study 1 and Study 2.**

|  | Study | b | p | Exp(B) | 95% CI Lower | 95% CI Upper |
|---|---|---|---|---|---|---|
| Age | S1 | −0.03 | 0.21 | 0.97 | 0.93 | 1.02 |
|  | S2 | 0.01 | 0.26 | 1.01 | 0.99 | 1.03 |
| Sex | S1 | 0.31 | 0.53 | 1.37 | 0.51 | 3.65 |
|  | S2 | 0.07 | 0.78 | 1.07 | 0.66 | 1.75 |
| Race | S1 | 0.22 | 0.69 | 1.25 | 0.42 | 3.72 |
|  | S2 | −0.19 | 0.58 | 0.83 | 0.42 | 1.63 |
| Education | S1 | 0.21 | 0.78 | 1.23 | 0.30 | 5.12 |
|  | S2 | −0.26 | 0.32 | 0.77 | 0.46 | 1.29 |
| Work industry | S1 | 0.62 | 0.40 | 1.85 | 0.45 | 7.68 |
|  | S2 | −0.10 | 0.80 | 0.91 | 0.43 | 1.91 |
| First language | S1 | NA | NA | NA | NA | NA |
|  | S2 | 0.86 | 0.08 | 2.36 | 0.91 | 6.13 |
| Antibiotics taken last year? | S1 | 0.70 | 0.21 | 2.02 | 0.67 | 6.07 |
|  | S2 | −0.73 | 0.03 | 0.48 | 0.25 | 0.92 |
| Doctor visited last year? | S1 | −0.42 | 0.46 | 0.66 | 0.22 | 1.98 |
|  | S2 | 0.78 | 0.00 | 2.17 | 1.27 | 3.73 |
| Vocabulary | S1 | 0.05 | 0.02 | 1.05 | 1.01 | 1.10 |
|  | S2 | −0.01 | 0.67 | 0.99 | 0.96 | 1.03 |
| Risk association | S1 | −0.38 | 0.10 | 0.69 | 0.44 | 1.07 |
|  | S2 | 0.21 | 0.03 | 1.24 | 1.02 | 1.50 |
| Concreteness | S1 | −0.10 | 0.55 | 0.90 | 0.65 | 1.26 |
|  | S2 | 0.03 | 0.60 | 1.04 | 0.91 | 1.18 |
| Familiarity | S1 | 0.27 | 0.24 | 1.31 | 0.84 | 2.06 |
|  | S2 | 0.27 | 0.00 | 1.31 | 1.09 | 1.59 |
| Processing fluency | S1 | −0.10 | 0.65 | 0.90 | 0.58 | 1.41 |
|  | S2 | −0.20 | 0.23 | 0.82 | 0.58 | 1.14 |
| Pronounceability | S1 | 0.25 | 0.27 | 1.28 | 0.83 | 1.99 |
|  | S2 | 0.22 | 0.21 | 1.25 | 0.88 | 1.79 |
| Constant | S1 | 0.82 | 0.66 | 2.27 |  |  |
|  | S2 | −0.51 | 0.66 | 0.60 |  |  |

The table shows betas values, significance levels, odds ratios and confidence intervals (CI) for adjusted odds ratios. Coefficients are from Block 3.

The study suffered from several methodological limitations. Participants completed the study online and in their own time and it is possible that they were distracted. Equally, they had the chance to search online for study terminology or look up test items contained within the WAIS vocabulary task. Indeed, a thorough analysis of WAIS responses indicated that some participants copied and pasted their answers from online contents (e.g., Wikipedia). This impact was mitigated by a very thorough data cleaning process, which involved manually identifying items copied and pasted, and scoring them with 0 points. An additional limitation pertains to potential confounds of some of our measures, notably risk association and memorability. Both risk perception and memorability of a term are likely to be influenced by a person's prior knowledge of it, and this needs to be considered when interpreting our results. However, since our correlation analyses indicated only weak to moderate correlations between familiarity ratings and ratings of risk perception and memorability respectively, we would argue that our use of these measures is still meaningful. Finally, a methodological shortcoming was identified with regard to the Likert rating scales employed for the linguistic attributes and risk association ratings. Our scales ranged from 1–7, with a middle point of "4" that was narratively described as "not sure". While the researchers intended the middle point to indicate a rating somewhere between "3" and "5", it is possible that participants assumed this rating to mean that they did not know the answer or did not have an opinion on the term in question. It is important to note this potential difference in interpretation, but since only 7.33% (Study 1) and 6.04% (Study 2) of all Likert ratings took on the middle value, we would argue that any confounds due to misinterpretation are negligible.

## Conclusions

Our findings highlight an urgent need to rename rather than merely "reframe" AMR in the public health domain. Coining different terminology that is memorable and effective in evoking perceptions of risk is likely going to be essential for improving the success of risk communication in the context of AMR. Identifying terms that are concrete, familiar, easy-to-process and pronounceable may help to increase risk perceptions in lay populations. However, a more complex interplay of demographic and individual difference variables (e.g., sex and vocabulary) may need to be considered when trying to increase memorability. Lessons might be learned from recent success stories including the renaming of the "Wuhan novel coronavirus" into "COVID-19".

## Data availability

The quantitative survey data that support the findings of this study are available in the Open Science Framework (OSF) with the identifier https://osf.io/4eb6j/. The source data underlying Figs. 2 and 3 is in Supplementary Data 2 and 3, respectively.

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

## Acknowledgements
This research was funded through a Leicester-Wellcome Trust ISSF Grant [Reference 204801/Z/16/Z].

## Author contributions
E.M.K.: Conceptualisation, Data collection, Data analysis, Writing – Original Draft, Writing – Review & Editing. K.O.C.: Data collection, Writing– Review & Editing. J.M.: Writing– Review & Editing. E.M.: Data analysis, Writing– Review & Editing.

## Competing interests
The authors declare no competing interests.
