## [Peer Review File · Communications Medicine]

Reviewers' comments:

Reviewer #1 (Remarks to the Author):

In this manuscript, the authors compare different terms used for antibiotic resistance with respect to their risk perception, memorability, and linguistic attributes. In two studies, people from the US (n = 267) and the UK (n = 965) rated 40 health related terms, including six AMR related terms, on risk perception and linguistic attributes. After completing a covariate test of crystallized intelligence (verbal intelligence from the WAIS), participants completed an active and passive memory test regarding the 40 health related terms. The authors find that the terms "AMR" and "Antimicrobial resistance" perform worse on almost all criteria than the better performing terms "drug-resistant infections" or "antibiotic resistance". In a follow-up analysis on the term "drug-resistant infections", the authors correlate term ratings with other term ratings, demographic variables and crystallized intelligence and have all these variables predict risk perception or the memorability indicators in step-wise regressions. They find that beyond demographic predictors, the linguistic dimensions and verbal intelligence predict risk perception and less systematic patterns for the memorability indices. The authors conclude that AMR should be reframed by using different terminology than established.

I believe the manuscript makes an important contribution to the literature. Not having an AMR background, the term AMR was weird to me, too, in the beginning, although I got used to it. However, the authors demonstrate convincingly, that there are better alternatives. Additionally, the manuscript is well written. However, next to some minor issues I will list at the end, I see major issues with the last part of the analyses.

First, the correlational and regression analyses are not well embedded in the rest of the manuscript. The choice of predictors in these models are not well based on literature reported in the introduction. Furthermore, there is a - for the reader - confusing mix of dependent and independent variables; i.e., risk perception and memorability scores are sometimes DV, sometimes IV, and linguistic attributes are IVs in the regression, but crucial criteria in the earlier analyses on term qualities across all terms. I believe the rationale for the regressions and also the analyses themselves have to be adjusted thoroughly. I would also recommend to drop the active memory as DV and IV, which would reduce the amount of results needed for these analyses. For arguments on this see below in the minor issues section.

Second, the argument given why the authors chose "drug-resistant infections" as the only term for the regression analyses is not convincing. The given argument is "others propose it as alternative", which might be a sufficient argument when empirical evidence is lacking. However, before the regression analyses, the authors themselves present empirical evidence on different terms. There, one conclusion is that "drug-resistant infections" is not the only superior term compared to "AMR" or "Antimicrobial resistance". If all criteria are weighted equally, one would have to conclude that "Antibiotic resistance" is the best term, so I still wonder, why this term was not (also) considered for the regression analyses. The authors either need to present more compelling arguments as to why they only focus on "drug-resistant infections" or they must at least also add "Antibiotic resistance". Overall, I'd recommend to run the same analyses for all AMR terms and choose only some for the manuscript and upload the others in online supplementary material such as OSF.

Additionally, there are minor issues that must be addressed:

1. P7L165f: Research question 3 is very vague and too broad to be answered by the studies. Given what can be answered by the studies, it also seems redundant with research question 1. Therefore, 3 should be revised or dropped.
2. P8L174: In my experience, your exclusion rates are extremely low for MTurk or Prolific samples. So you either have prescreened better than others do or you did not clean your data sufficiently. Please add further information on sampling and cleaning. If the sampling was not superior to other studies, then I'd expect that there is still many invalid responders in the data. If so, you might want to consider additional measures to get your data clean, such as checking for invariant or pattern responding or excluding participants whose scores are oftentimes outliers.

3. P9L188: Why does the US sample have a that much higher SD in the WAIS score? Please provide further information.
4. P10L208: It is much appreciated that you control for general mental abilities for your research, but please adhere to contemporary intelligence terms in your construct and skill labels. I.e., there is no such thing as "verbal intelligence". Even the heavily outdated and always decades late WAIS has finally adopted gf/gc terminology and the subtest you use is clearly gc. The skill assessed there is not "reading skill", but vocabulary (Stratum I) as an indicator of crystallized intelligence (gc, Stratum II).
5. P12L245ff: By your design, the passive memory test is stochastically dependent on the active memory test. To keep it short, this leaves the passive test fully uninterpretable. This becomes more obvious when inspecting results related to the passive test. Any results related to this test must be excluded/rerun without it. Also, this might change some of your interpretations and conclusions in the discussion.
6. P12L266ff: How many raters rated the WAIS responses? There should be at least two. With two or more raters, please report interrater reliabilities.
7. P13L280: The inspection of scatter plots is good, but not a "test" of normality or for outliers. More standardized approaches would be excluding univariate (e.g. outside the $\pm 3SD$ interval) and multivariate (outside the 95% percentile of the Mahalanobis distance χ^2 -distribution) outliers.
8. P13L283: If I understand it correctly, the regressions you run are step-wise. This is a better term than hierarchical, because it is less confusing (hierarchical can be confused with multilevel).
9. P14L300ff: For risk perception, people must know a term. Did you measure, manipulate, or control for that in any way? If not, it is easily explained why AMR probably has the lowest score here. Simply because lay people probably don't know the term. That's still a problem with the term, but then familiarity would be driving the risk perception score, which means the score does not (purely) reflect risk perception, but something else.
10. P16L324ff: To strengthen this section, you should consider adding post-hoc t-tests of the highest and second highest terms.
11. P18L364: It is striking, that "drug-resistant infections" has so low memorability scores. However, it might be explained by term not being used widely. Thus, your memorability score is not purely a memorability score, but also reflects how well known or established the term is in a society. This contagion of the score must be discussed and considered in your interpretations.
12. P23f correlations matrices: First of all, I do not find these tables very informative and they are also not embedded well. This might change while you answer my major concern with the manuscript, but if not, please work on embedding them better or move them to the supplement. Second, if you keep the matrices, please combine to one table. I.e., have one study in the upper triangle and the other in the lower triangle. This saves space and makes comparisons easier. Third, in the second table there probably is a typo in l3c2, I guess the value should be .054 not 0.54, right?
13. P25L439ff: please report CIs for standardized regression weights. With psychological variables that do not have natural scales, unstandardized values are not interpretable.
14. P571ff: For future studies with tests can be solved via quick google searches, you might want to consider using PageFocus (Diedenhofen & Musch, 2017). Really useful tool! Diedenhofen, B., & Musch, J. (2017). PageFocus: Using paradata to detect and prevent cheating on online achievement tests. *Behavior Research Methods*, 49, 1444-1459. <https://doi.org/10.3758/s13428-016-0800-7>

Reviewer #2 (Remarks to the Author):

In this research, the author/s tested whether various terms describing antimicrobial resistance propel different psychological characteristics (e.g., subjective riskiness, memorability, pronounceability).

They conducted two studies to discover that even though the term “drug-resistant infections” is perceived as conveying the highest risk among the tested terms, it does not have very good memorability. They concluded that we urgently need to rename antimicrobial resistance with a new word.

I think that this is an important research avenue and agree with the authors that the terminology used for public health communication should be tested. The authors conducted the studies diligently and the data were adequately analysed.

I am happy to recommend publication, but I do have a few concerns that should be addressed with additional analyses and in the limitation section of the Discussion.

First, the authors should clarify in the paper that the default term that needs changing is antimicrobial resistance/AMR, not the other commonly used terms (if this was the case). Otherwise, I am not sure what establishes the urgent need to change the terms. For instance, looking at the data patterns, the term “antibiotic resistance” is performing pretty well in most measured dimensions. In fact, even in the riskiness assessment, there is only a small difference between the “drug-resistant infections” and the “antibiotic resistance” terms. Should not the term antibiotic resistance be recommended for communication based on this research?

Second, I believe understanding subjective risk perception is essential. But in this instance, it would also be good to know how well the different terms are calibrated in respect with the objective risk value. I assume risk is here intended to mean the condition’s severity (e.g., number of deaths due to the condition) combined with its probability of occurrence. I think some additional comparisons can be made in this respect as we can attach, for example, the number of deaths and multiply with their probability of contracting them to most of the conditions in the list. Such numbers can be compared with the AMR terms. At the very least, the lack of calibration of each term with respect to the objective riskiness value should be discussed in the limitations section.

Third, I was not sure about how participants interpreted the middle point of the scales (“not sure It can be interpreted as meaning “I don’t know how risky”, in which case it should not be included in the mean calculations or as meaning “somewhere in between somewhat risky or somewhat not risky” in which case mean calculation would be appropriate. Would the mean results be similar to those presented in the manuscript if this point is excluded? ”). I was also not sure how common these levels were: we don’t see the distribution of the answers per each response level. I would like the authors to discuss this as a possible limitation in the Discussion section and, if possible, summarise the additional analysis in the Supplementary Materials.

Four, I have some additional minor points:

Perhaps a more appropriate description of the sampling method used for Prolific should be a quota representative sample or a nationally representative sample using the quota of age and sex if I understand correctly how Prolific implements the quota sampling.

Please specify whether the dependent measures were presented in a random or fixed order to participants. Please report the randomisation procedure if relevant, including the randomisation procedure for the terms randomisation.

The authors claim that no research has empirically tested the terms' effectiveness; it seems that some qualitative analyses were reported in the 2019 Wellcome Trust report.

I understand that the authors did not want to overwhelm the readers with statistical tests. The statistical tests for all pairwise comparisons conducted and narratively reported in the main text, however should be reported in full at least in the Supplementary Materials. I believe that correction for

multiple testing should also be used to avoid Alpha inflation (it was not clear to me this was done).

The authors could make their data publicly available.

Overall, this research represents an important research avenue that deserves to be published in a high-profile journal. Some questions and limitations should be addressed in a revision.

Dear Dr Rawson,

Thank you for considering our recent submission to Communications Medicine and for your invitation to revise and resubmit our article.

We have reviewed the paper and made significant changes in line with the suggestions of the three reviewers. Below follows our detailed response to individual reviewer comments.

Yours sincerely,
Eva Krockow

Reviewer #1:

In this manuscript, the authors compare different terms used for antibiotic resistance with respect to their risk perception, memorability, and linguistic attributes. In two studies, people from the US (n = 267) and the UK (n = 965) rated 40 health related terms, including six AMR related terms, on risk perception and linguistic attributes. After completing a covariate test of crystallized intelligence (verbal intelligence from the WAIS), participants completed an active and passive memory test regarding the 40 health related terms. The authors find that the terms “AMR” and “Antimicrobial resistance” perform worse on almost all criteria than the better performing terms “drug-resistant infections” or “antibiotic resistance”. In a follow-up analysis on the term “drug-resistant infections”, the authors correlate term ratings with other term ratings, demographic variables and crystallized intelligence and have all these variables predict risk perception or the memorability indicators in step-wise regressions. They find that beyond demographic predictors, the linguistic dimensions and verbal intelligence predict risk perception and less systematic patterns for the memorability indices. The authors conclude that AMR should be reframed by using different terminology than established. I believe the manuscript makes an important contribution to the literature. Not having an AMR background, the term AMR was weird to me, too, in the beginning, although I got used to it. However, the authors demonstrate convincingly, that there are better alternatives. Additionally, the manuscript is well written. However, next to some minor issues I will list at the end, I see major issues with the last part of the analyses.

1. First, the correlational and regression analyses are not well embedded in the rest of the manuscript. The choice of predictors in these models are not well based on literature reported in the introduction. Furthermore, there is a - for the reader - confusing mix of dependent and independent variables; i.e., risk perception and memorability scores are sometimes DV, sometimes IV, and linguistic attributes are IVs in the regression, but crucial criteria in the earlier analyses on term qualities across all terms. I believe the rationale for the regressions and also the analyses themselves have to be adjusted thoroughly. I would also recommend to drop the active memory as DV and IV, which would reduce the amount of results needed for these analyses. For arguments on this see below in the minor issues section.

Response: *Many thanks for these comments. We agree that the correlational and regression analyses needed better embedding. We have done this by moving the correlation tables to the supplementary materials (also see our response to comment 14) and adapting the introduction to the Results sub-section on identifying predictor variables as follows: “Following our comparison of word effectiveness for the six AMR-related health terms, we aim to identify significant predictors of word effectiveness. In our analyses, we focus on two of the AMR-related health terms; “Drug-resistant infections” and “Antibiotic resistance”. The former was chosen due to the term’s support by previous literature (Mendelson et al., 2017; Wellcome Trust, 2019) and because it achieved the highest risk association ratings across*

both our studies. The latter was chosen, because the term performed best across both word effectiveness measures in Studies 1 and 2."

We respectfully disagree with regard to the choice of model predictors. We believe that these are fully justified based on our detailed review of linguistic factors predicting word attensity in the Introduction. The choice of demographic control variables is explained in the Design section.

We thank the reviewer for highlighting the confusing labelling of variables in the previous manuscript version. To avoid confusion about the role of our linguistic attributes (concreteness, familiarity, fluency and pronounceability), we have removed the previous part of the Results section, which treated the attributes as DVs as part of a statistical comparison across all AMR-related health terms. By deleting this part, we ensure that linguistic attributes are consistently treated as IVs/predictor variables and only feature in the later correlation and regression analyses. As for risk perception and memorability, we have checked the Design section and ensured that these are consistently labelled as dependent variables (both in the ANOVAs and subsequent regressions).

Finally, we followed the reviewer's recommendation to drop active memory as DV and IV (see also response to Comment 7).

2. Second, the argument given why the authors chose "drug-resistant infections" as the only term for the regression analyses is not convincing. The given argument is "others propose it as alternative", which might be a sufficient argument when empirical evidence is lacking. However, before the regression analyses, the authors themselves present empirical evidence on different terms. There, one conclusion is that "drug-resistant infections" is not the only superior term compared to "AMR" or "Antimicrobial resistance". If all criteria are weighted equally, one would have to conclude that "Antibiotic resistance" is the best term, so I still wonder, why this term was not (also) considered for the regression analyses. The authors either need to present more compelling arguments as to why they only focus on "drug-resistant infections" or they must at least also add "Antibiotic resistance". Overall, I'd recommend to run the same analyses for all AMR terms and choose only some for the manuscript and upload the others in online supplementary material such as OSF.

Response: *This is a fair comment. Our choice of the term "drug-resistant infections" was driven mostly by previous expert recommendations, but we agree that this does not constitute a compelling rationale. We therefore followed the reviewer's suggestion to also run detailed analyses for the term "antibiotic resistance", which indeed outperformed other AMR-related terms in terms of word effectiveness. We have consequently extended our previous Results section to include regression analyses for "antibiotic resistance". Additionally, we have revised the research questions to drop the singular focus on "drug-resistant infections" and adapted our Introduction and Discussion accordingly.*

Additionally, there are minor issues that must be addressed:

3. P7L165f: Research question 3 is very vague and too broad to be answered by the studies. Given what can be answered by the studies, it also seems redundant with research question 1. Therefore, 3 should be revised or dropped.

Response: *This is a fair point. We have dropped research question 3 as suggested.*

4. P8L174: In my experience, your exclusion rates are extremely low for MTurk or Prolific samples. So you either have prescreened better than others do or you did not clean your data sufficiently. Please

add further information on sampling and cleaning. If the sampling was not superior to other studies, then I'd expect that there is still many invalid responders in the data. If so, you might want to consider additional measures to get your data clean, such as checking for invariant or pattern responding or excluding participants whose scores are oftentimes outliers.

Response: We thank the reviewer for this comment and added an additional check for pattern responding. Specifically, we screened for straightlining (i.e., giving the same score or very similar scores across items). For each term, the standard deviation of Likert scores was calculated across the items for concreteness, familiarity, fluency, pronounceability and risk perception. Subsequently, we calculated the mean standard deviation across all terms. Individuals with scores of $\leq .04$ were excluded on suspicion of straightlining. As a result of this additional step of data screening, we excluded 14 more participants from Study 1 and 40 more participants from Study 2.

Additionally, we included a check of engagement in the WAIS vocabulary task (see response Comment 5 below). This resulted in a further 16 participants being excluded from Study 1 and 1 participant excluded from Study 2. All reporting and analyses have been adapted based on the new sample sizes.

5. P9L188: Why does the US sample have a that much higher SD in the WAIS score? Please provide further information.

Response: We examined the distributions of WAIS scores for the two samples in more detail. Histograms indicated that the US sample had a much higher proportion of very low scores, which increased overall variability in the data. We considered the issue of different SDs carefully. We do not believe that the different SDs present a problem per se, because it is possible that the US sample had a broader range of knowledge in the context of vocabulary. However, the high number of very low scores did cause concern, because even individuals with only a rudimentary grasp of the English language should be able to provide some definition of the first few test items ("apple", "glove", "breakfast" and "curious"). We therefore decided that a very low WAIS score of < 5 was an indicator of either 1) very poor English language skills or 2) very poor study engagement, both of which we considered additional grounds for exclusion from the study. As a consequence, we ended up adding the WAIS score as another data screening criterion and deleted another 16 participants from Study 1 and 1 participant from Study 2, who failed to achieve a high enough WAIS score. In addition to hopefully increasing overall data quality, this had the result of reducing the WAIS SD for the US sample to 11.09, which is still higher than the UK SD, but lower than the previous value.

6. P10L208: It is much appreciated that you control for general mental abilities for your research, but please adhere to contemporary intelligence terms in your construct and skill labels. I.e., there is no such thing as "verbal intelligence". Even the heavily outdated and always decades late WAIS has finally adopted gf/gc terminology and the subtest you use is clearly gc. The skill assessed there is not "reading skill", but vocabulary (Stratum I) as an indicator of crystallized intelligence (gc, Stratum II).

Response: We are grateful for this correction. It is true that verbal intelligence is a misleading term. We used it because it was included in previous versions of the WAIS, but we agree that the verbal subtest used in our study constitutes a test of vocabulary and falls within the category of crystallised intelligence. We have therefore revised our wording throughout the manuscript (e.g., replacing "verbal intelligence" with "vocabulary").

7. P12L245ff: By your design, the passive memory test is stochastically dependent on the active memory test. To keep it short, this leaves the passive test fully uninterpretable. This becomes more obvious when inspecting results related to the passive test. Any results related to this test must be

excluded/rerun without it. Also, this might change some of your interpretations and conclusions in the discussion.

Response: *We thank the reviewer for this critical comment. Based on the arguments presented here and the recommendation made in the first comment, we have decided to drop the active memory variable throughout.*

8. P12L266ff: How many raters rated the WAIS responses? There should be at least two. With two or more raters, please report interrater reliabilities.

Response: *One primary rater (KC) and one secondary rater (EK) rated the WAIS responses. No formal interrater reliabilities were calculated, but a thorough process was followed, whereby the primary rater flagged all ambiguous items (approximately 10%) and the secondary rater checked the items in question. The two raters discussed any items of disagreement until they reached a consensus.*

9. P13L280: The inspection of scatter plots is good, but not a "test" of normality or for outliers. More standardized approaches would be excluding univariate (e.g. outside the +3SD interval) and multivariate (outside the 95% percentile of the Mahalanobis distance χ^2 -distribution) outliers.

Response: *This is a fair point. However, given the fairly large sample sizes and the general robustness of ANOVAs to minor violations of the normality assumptions, we decided that an inspection of scatter plots was sufficient and appropriate in this instance.*

10. P13L283: If I understand it correctly, the regressions you run are step-wise. This is a better term than hierarchical, because it is less confusing (hierarchical can be confused with multilevel).

Response: *We respectfully disagree and believe that "hierarchical regression" is the correct term. It refers to the process by which one introduces predictors based on hypothesised importance (least important go in block 1, most important go in final block). Stepwise is a data-driven approach that includes only significant variables, and drops non-significant from the model. We have therefore decided to retain the term "hierarchical".*

11. P14L300ff: For risk perception, people must know a term. Did you measure, manipulate, or control for that in any way? If not, it is easily explained why AMR probably has the lowest score here. Simply because lay people probably don't know the term. That's still a problem with the term, but then familiarity would be driving the risk perception score, which means the score does not (purely) reflect risk perception, but something else.

Response: *We agree that societal use of and subsequent knowledge of a term can contribute to risk perception, but we would respectfully disagree that this is a pre-requisite. For example, it may not be necessary for an individual to have heard of terms such as 'drug-resistant infections' or 'superbugs' to sense a danger or harm associated with them. Indeed, our correlation analyses showed that risk association was significantly correlated with familiarity for the term 'drug-resistant infections', but with an r -value of $<.3$, this suggested a comparatively weak relationship.*

We had considered controlling for confounds by testing participants' existing knowledge of different AMR terms prior to completion of the study's main rating task. However, we decided against this approach, because we thought that assessing knowledge about the terms would inflate later familiarity ratings and memory scores.

We agree that it is difficult to measure pure risk perception (without the influences of any confounds such as familiarity), but we would argue that our approach is still meaningful. Nevertheless, we have added the follow sentences about potential confounds to our section on study limitations: "An additional limitation pertains to potential confounds of some of our

measures, notably risk association and memorability. Both risk perception and memorability of a term are likely to be influenced by a person's prior knowledge of it, and this needs to be considered when interpreting our results. However, since our correlation analyses indicated only weak to moderate correlations between familiarity ratings and ratings of risk perception and memorability respectively, we would argue that our use of these measures is still meaningful."

12. P16L324ff: To strengthen this section, you should consider adding post-hoc t-tests of the highest and second highest terms.

Response: *We thank the reviewer for this suggestion. We decided to include detailed tables with post-hoc tests comparing all terms in the supplementary materials.*

13. P18L364: It is striking, that "drug-resistant infections" has so low memorability scores. However, it might be explained by term not being used widely. Thus, your memorability score is not purely a memorability score, but also reflects how well known or established the term is in a society. This contagion of the score must be discussed and considered in your interpretations.

Response: *We believe that this comment addresses a similar criticism as comment #11. Please see how response above and our addition to the section on study limitations.*

14. P23f correlations matrices: First of all, I do not find these tables very informative and they are also not embedded well. This might change while you answer my major concern with the manuscript, but if not, please work on embedding them better or move them to the supplement. Second, if you keep the matrices, please combine to one table. I.e., have one study in the upper triangle and the other in the lower triangle. This saves space and makes comparisons easier. Third, in the second table there probably is a typo in l3c2, I guess the value should be .054 not 0.54, right?

Response: *We agree that these tables were not essential to the main manuscript and could be condensed by combining the results from both studies within a single table. We have therefore moved the reformatted correlation tables to the supplementary materials. Many thanks for spotting the typo. We updated and checked all table values following our more stringent screening of both data sets.*

15. P25L439ff: please report CIs for standardized regression weights. With psychological variables that do not have natural scales, unstandardized values are not interpretable.

Response: *Many thanks for this suggestion. We have replaced the previously reported CIs for unstandardised regression weights with approximate CIs for standardised regression weights that were obtained by converting the scores of our variables into z scores and re-running the regression analyses, using bootstrapping (N = 2000).*

16. P571ff: For future studies with tests can be solved via quick google searchers, you might want to consider using PageFocus (Diedenhofen & Musch, 2017). Really useful tool! Diedenhofen, B., & Musch, J. (2017). PageFocus: Using paradata to detect and prevent cheating on online achievement tests. *Behavior Research Methods*, 49, 1444-1459. <https://doi.org/10.3758/s13428-016-0800-7>

Response: *Thank you very much for highlighting this resource! We were not aware this tool existed and will definitely consider using it in future studies.*

Reviewer #2:

In this research, the author/s tested whether various terms describing antimicrobial resistance propel different psychological characteristics (e.g., subjective riskiness, memorability, pronounceability). They conducted two studies to discover that even though the term "drug-

resistant infections” is perceived as conveying the highest risk among the tested terms, it does not have very good memorability. They concluded that we urgently need to rename antimicrobial resistance with a new word.

I think that this is an important research avenue and agree with the authors that the terminology used for public health communication should be tested. The authors conducted the studies diligently and the data were adequately analysed.

I am happy to recommend publication, but I do have a few concerns that should be addressed with additional analyses and in the limitation section of the Discussion.

1. First, the authors should clarify in the paper that the default term that needs changing is antimicrobial resistance/AMR, not the other commonly used terms (if this was the case). Otherwise, I am not sure what establishes the urgent need to change the terms. For instance, looking at the data patterns, the term “antibiotic resistance” is performing pretty well in most measured dimensions. In fact, even in the riskiness assessment, there is only a small difference between the “drug-resistant infections” and the “antibiotic resistance” terms. Should not the term antibiotic resistance be recommended for communication based on this research?

Response: *This is an interesting point. We would argue that it isn't one specific term that needs changing, but rather the current approach of using six or more terms inconsistently, with some of them (notably “antimicrobial resistance” and “AMR”) appearing particularly unsuitable. To increase clarity of our aims, we added the following sentence to the Introduction under “Study aims and research questions”:*

“Rather than replacing a single AMR-related term, we seek to change the current use of inconsistent language by testing whether any of the existing terminology has the potential to serve as uniform, key term.” We also added the following sentence to the first paragraph of the Discussion: “We aimed to test whether a single existing term could replace the use of inconsistent terminology, which is typical of current health risk communication and media reporting.”

We agree that “antibiotic resistance” performed well compared with other AMR-related health terms. In fact, following the reviewer’s comments, we decided to include the term “antibiotic resistance” in the more detailed regression analyses to identify relevant predictor variables for its word effectiveness. We also added a more explicit comparison of “drug-resistant infections” and “antibiotic resistance” to our Discussion section:

“However, the results of both studies also showed that “Drug-resistant infections” ranked particularly low on memorability, while “Antibiotic resistance” was remembered most easily. These findings suggest that “Drug-resistant infections” surpasses the other existing terms only on one dimension of word effectiveness, while “Antibiotic resistance” appears to be the most effective AMR-related health term overall.”

Following our analyses, however, we still have doubts about the overall effectiveness of any one existing term, given the low ratings compared with terminology pertaining to other major health risks, many of which are objectively less deadly (see also our response to Comment 2). We have amended our Discussion section to provide a clearer justification under “Implications for Policy and Practice”: “The combined results from our two survey studies in the US and UK suggest that no existing AMR-related health term is likely to be sufficient for drawing the warranted attention to AMR's global public health issue. “AMR” and “Antimicrobial resistance” appear particularly unsuitable for public health risk communication, but it is questionable whether other existing terms can offer effective alternatives. The term “Drug-resistant infections”, which was recommended to replace all

other AMR-related health terms in future communications (Mendelson et al., 2017; Wellcome Trust, 2019), appears to have some advantages due to its ability to evoke higher risk perceptions, but—notably—these risk perceptions still scored lower than other health risk and disease terms with similar death tolls (e.g., obesity). Additionally, its low memorability questions the term’s overall effectiveness. Indeed, if considering both risk association and memorability ratings, the term “Antibiotic resistance” could be considered superior to “Drug-resistant infections”. Yet again, “Antibiotic resistance” falls short in its effectiveness compared to other major health risk terms.”

2. Second, I believe understanding subjective risk perception is essential. But in this instance, it would also be good to know how well the different terms are calibrated in respect with the objective risk value. I assume risk is here intended to mean the condition’s severity (e.g., number of deaths due to the condition) combined with its probability of occurrence. I think some additional comparisons can be made in this respect as we can attach, for example, the number of deaths and multiply with their probability of contracting them to most of the conditions in the list. Such numbers can be compared with the AMR terms. At the very least, the lack of calibration of each term with respect to the objective riskiness value should be discussed in the limitations section.

Response: *We appreciate this comment. Risk perception is an ambiguous concept. In the health psychology literature (notably the Health Belief Model), it is typically considered to entail beliefs about both a condition’s severity and personal susceptibility or vulnerability to the risk in question. We have made this clearer in the Introduction by modifying an existing sentence in the second paragraph following the sub-heading “Measuring the effectiveness of terminology”: “Models of health behaviour, such as Rosenstock’s (1974) seminal Health Belief Model, highlight the important role of risk perception, which entails beliefs about both a risk’s severity and personal susceptibility or vulnerability to it, (i.e. the recognition of susceptibility to a severe threat) in motivating certain health-related behaviours.”*

We liked the idea of comparing actual, objective risk with participants’ subjective perceptions to better judge participants’ calibration levels. Consequently, we have created a table containing information about the 2019 mortality rates associated with each health risk term included in the study and contrasted this information with the mean risk association ratings from Study 1 and Study 2. Due to space constraints, we have added the table to our supplementary materials, but we have included the following discussion in the second paragraph of our Discussion section:

“Consulting existing mortality data reported for all 40 health terms (see Table 7 in supplementary materials), which might be considered as a proxy for objective health risk, it becomes evident that the general public’s risk perceptions are miscalibrated. While participants correctly judged heart disease and cancer to be amongst the largest health threats, they severely overestimated the risks of tropical diseases such as Ebola and Malaria, while underestimating the threat of AMR, which ranked 6th in terms of global deaths incurred and is predicted to overtake cancer as a leading cause of death by 2050 (O’Neill, 2016).” We also referred to the table in our Materials section, when justifying the selection of the 40 health risk and disease terms used across our surveys.

3. Third, I was not sure about how participants interpreted the middle point of the scales (“not sure It can be interpreted as meaning “I don’t know how risky”, in which case it should not be included in the mean calculations or as meaning “somewhere in between somewhat risky or somewhat not risky” in which case mean calculation would be appropriate. Would the mean results be similar to those presented in the manuscript if this point is excluded? ”). I was also not sure how common these levels were: we don’t see the distribution of the answers per each response level. I would like

the authors to discuss this as a possible limitation in the Discussion section and, if possible, summarise the additional analysis in the Supplementary Materials.

Response: *Many thanks for this comment. We had not considered the possibility of different interpretations of our scales' middle points. Our assumption was that participants would interpret them to mean "somewhere between somewhat risky and somewhat not risky", which is why we included the mean in our calculations. Following the reviewer comment, we have given this issue additional consideration and added the following paragraph to our Limitations section:*

"Finally, a methodological shortcoming was identified with regard to the Likert rating scales employed for the linguistic attributes and risk association ratings. Our scales ranged from 1-7, with a middle point of "4" that was narratively described as "not sure". While the researchers intended the middle point to indicate a rating somewhere between "3" and "5", it is possible that participants assumed this rating to mean that they did not know the answer or did not have an opinion on the term in question. It is important to note this potential difference in interpretation, but since only 7.33% (Study 1) and 6.04% (Study 2) of all Likert ratings took on the middle value, we would argue that any confounds due to misinterpretation are negligible."

Four, I have some additional minor points:

4. Perhaps a more appropriate description of the sampling method used for Prolific should be a quota representative sample or a nationally representative sample using the quota of age and sex if I understand correctly how Prolific implements the quota sampling.

Response: *Thank you for the suggestion. We have double-checked the information about Prolific's approach to nationally representative sampling as provided on their website (<https://researcher-help.prolific.co/hc/en-gb/articles/360019236753-Representative-samples>) and have clarified the sampling method in our participants section as follows: "For Study 2, UK participants were recruited as a nationally representative sample (using the UK Office of National Statistics' quota of age, sex and ethnicity) through the online platform Prolific in November 2021."*

5. Please specify whether the dependent measures were presented in a random or fixed order to participants. Please report the randomisation procedure if relevant, including the randomisation procedure for the terms randomisation.

Response: *The dependent measures were presented in fixed order. While we appreciate that randomisation could have prevented potential order effects, we thought that a fixed order would help to avoid confusion amongst participants. The randomisation procedure for the display of health terms differed between the two studies. For Study 1, technical constraints meant that full randomisation was not possible. We therefore created 10 different survey versions based on 10 different word orders that were produced using the Rand() function in Excel. Equal numbers of participants were allocated to each survey version. For Study 2, Gorilla software, using Javascript, applied the shuffle function to create pseudorandom trial orders, using a different "seed" for each participant. We have adjusted information in the Methods section (under Design and Materials respectively) to provide more detailed information about our approach.*

6. The authors claim that no research has empirically tested the terms' effectiveness; it seems that some qualitative analyses were reported in the 2019 Wellcome Trust report.

Response: *This is a fair point. We have added a clarifying statement in the Introduction: "Despite academic support for the term "Drug-resistant infections", no research has*

conducted a comprehensive empirical test of its effectiveness or compared it to the other existing AMR-related terms. Some qualitative results have highlighted the overuse of technical jargon in the context of AMR communication and indicated lay people's difficulties in making sense of terminology relating to microbes (Wellcome Trust, 2019), but there is an urgent need for large-scale, quantitative research on the effectiveness of AMR terminology."

7. I understand that the authors did not want to overwhelm the readers with statistical tests. The statistical tests for all pairwise comparisons conducted and narratively reported in the main text, however should be reported in full at least in the Supplementary Materials. I believe that correction for multiple testing should also be used to avoid Alpha inflation (it was not clear to me this was done).

Response: *This is a fair comment. We have now included pairwise comparisons with Bonferroni adjustments for multiple testing for all ANOVAs in the supplementary materials.*

8. The authors could make their data publicly available.

Response: *Thank you for the suggestion. The data are now publicly available on the OSF and we have added the URL (<https://osf.io/4eb6j/>) to our manuscript.*

9. Overall, this research represents an important research avenue that deserves to be published in a high-profile journal. Some questions and limitations should be addressed in a revision.

Response: *We thank the reviewer for this positive evaluation of our work and hope to have addressed all questions and limitations in our comments and revisions.*

Reviewer #3:

The manuscript presents two survey studies that examine perceptions (risk, memorability, and linguistic dimensions of concreteness, familiarity, processing fluency, and pronounceability) of the general public in the United Kingdom and United States of six terms related to antimicrobial resistance.

I have the following comments on the manuscript.

1. Both studies do not test any specific a priori expectations and their goal is to find out people's perceptions. The manuscript, overall, present a sound rationale for the studies.

Response: *We thank the reviewer for the overall positive evaluation of our work.*

2. Although not critical, it would be nice to have the six terms presented in table along with some very brief information about their history and context of use. At this time, it is not clear how and why these six terms were selected and the table can perhaps provide that rationale.

Response: *The six terms were selected, because they are the ones most frequently used in the media. We have clarified this in the Introduction. We thank the reviewer for the excellent suggestion of adding an overview table. This has now also been included in the Introduction (Table 1).*

3. Please provide some citations that support use of Amazon mTurk for sample recruitment and data quality of these samples.

Response: *We agree that it is important to discuss the quality of online data samples and have added the following information and additional citations to our Participants section: "Amazon MTurk was previously highlighted as an online data collection platform with a large participant pool, good diversity levels and convenient access to high-quality data*

(Buhrmester, Kwang & Gosling, 2016). Recent years (2018-2019) have seen rising concerns about increasing numbers of inattentive participants and so-called “bots” (Chmielewski & Kucker, 2020), and a sudden change during the start of the COVID-19 pandemic, when lockdowns and changes to the job market meant that many people turned to online alternatives to earn an income. While this change resulted in more people joining the MTurk participant pool and further increased sample representativeness, concerns regarding participant attentiveness persisted (Arechar & Rand, 2021). Recognising the potential of poor-quality data obtained through Amazon MTurk, we conducted stringent data quality screening (reported below) and implemented a follow-up survey recruited through Prolific, which has repeatedly been shown to produce higher quality data (Douglas, Ewell, Brauer, 2023).”

Following reviewer comments and additional consideration of data quality concerns, we decided to extend the data screening procedures reported in our previous manuscript version. Specifically, we conducted an additional check for pattern responding and screened for straightlining (i.e., giving the same score or very similar scores across items). We detailed our approach in the Data analysis section: “Finally, all Likert ratings for the 40 health terms were screened for straightlining. For each term, the standard deviation of Likert scores was calculated across the items concreteness, familiarity, fluency, pronounceability and risk perception. Subsequently, we calculated the mean standard deviation across all terms. Individuals with scores of $\leq .04$ were excluded on suspicion of straightlining.” As a result of this additional step of data screening, we excluded 14 more participants from Study 1 and 40 more participants from Study 2. All reporting and analyses have been adapted based on the new sample sizes. Additionally, we included a further check of engagement in the WAIS vocabulary task. This resulted in a further 16 participants being excluded from Study 1 and 1 participant excluded from Study 2. All reporting and analyses have been adapted based on the new sample sizes.

The following references were added to the reference list:

*Arechar, A.A., Rand, D.G. Turking in the time of COVID. Behav Res 53, 2591–2595 (2021).
<https://doi.org/10.3758/s13428-021-01588-4>*

*Buhrmester, M., Kwang, T., & Gosling, S. D. (2016). Amazon's Mechanical Turk: A new source of inexpensive, yet high-quality data? In A. E. Kazdin (Ed.), Methodological issues and strategies in clinical research (pp. 133–139). American Psychological Association.
<https://doi.org/10.1037/14805-009>*

Chmielewski, M., & Kucker, S. C. (2020). An MTurk crisis? Shifts in data quality and the impact on study results. Social Psychological and Personality Science, 11(4), 464-473.

Douglas BD, Ewell PJ, Brauer M (2023) Data quality in online human-subjects research: Comparisons between MTurk, Prolific, CloudResearch, Qualtrics, and SONA. PLoS ONE 18(3): e0279720. <https://doi.org/10.1371/journal.pone.0279720>

4. I would have liked to have seen the actual survey completion times presented (instead of the 10-20 minutes estimate) for each study. Looking at the length of the questionnaire and the number of questions, I would have thought that the actual mean time for completion would be longer.

Response: *We have considered this comment, but decided against including completion times, because it is difficult to estimate the exact times participants spent completing the study. While we have data about the time they started the study and the time they reached*

the final page, real-time monitoring showed that many participants took breaks during the survey and didn't complete it in one sitting. Many participants took up to one hour to complete the study (which was the maximum amount of time given before being timed out by the system), but we think reporting these times would be misleading, because they are unlikely to reflect the actual time spent on the study. We provided the 10-20 estimate based on test runs conducted prior to launching the study.

5. Are there any concerns about family wise error rates for statistical analyses? For example, there are several pairwise comparisons for several analyses - should the p value for these comparisons be adjusted?

Response: *This is a fair comment. To increase overall transparency of our pairwise comparisons, we have provided detailed tables for all our ANOVAs in the supplementary materials. To account for multiple comparisons and potential alpha inflation, we have used Bonferroni adjustments throughout.*

6. The manuscript will be most strengthened by a better organization of the discussion section. Typically, this section has the following subsections: summary and explanation of results in light of the theoretical framework and research problem, implications of results for theory (which may not be needed for this manuscript), implications for practice and application, suggestions for future research, limitations of the study, and conclusions.

Response: *This is a helpful suggestion. We have strengthened the previous structure of our Discussion by adding a new section on "implications for practice and policy", a brief section on "future research directions" and a separate sub-heading for our conclusions.*

7. Overall, the manuscript makes an important contribution to the design of effective public health messages related to antimicrobial resistance.

Response: *We thank the reviewer for this positive feedback about our manuscript.*

REVIEWERS' COMMENTS:

Reviewer #1 (Remarks to the Author):

I want to thank the authors for their thorough revisions and also their rebuttals. I believe the manuscript has improved enormously and will be a great addition to the literature and help AMR researchers to use the most appropriate terminology in their studies or when conducting science communication to broader audiences.

I am content with all changes that were made and recommend publication of the manuscript. I just have one final comment that the authors might or might not consider. With regards to my initial 10th comment about the type of regression, I am aware that in some disciplines the terminology also used in this manuscript is standard. From my experience, this stems from a misinterpretation of step-wise regression, which in these disciplines is considered only a data driven optimization method which might sometimes be understood as data tweaking. But from a pure methodological standpoint, step-wise just means computing nested regression models one after the other and comparing their delta R^2 , and the step-wise regression optimization algorithms are just one of many many ways to use step-wise regressions. Given that hierarchical clearly sometimes (in more methodological disciplines) is confused with multilevel modeling, I still recommend to change the term. However, considering different traditions between disciplines, this is optional I guess.

Reviewer #2 (Remarks to the Author):

Thank you to the authors for carefully revising the paper. They have successfully addressed most of my concerns and comments. I am happy to endorse the publication pending some minor revisions. I have only a few minor issues that I kindly ask the authors to address.

Please modify the sentence in the abstract: "The necessary behaviour change relies on effective public health communication, but previous information campaigns and media coverage have failed to raise public awareness." I don't think this is a true statement given the existing evidence (see Huttner, 2010 and Thoolen, 2012 for a narrative review and meta-analysis) of the effect of public campaigns on improving knowledge and attitudes towards AMR. However, the effectiveness of the campaign could surely be improved.

I recommend avoiding the term "word effectiveness" in the abstract. The term is not self-explanatory for non-linguists and could be confusing. While it is well-defined in the paper, readers who only read the abstract may not understand it. I suggest replacing the term in the abstract with more self-explanatory terms like "memorability" and "risk association."

The OSF link was created to share the data, but currently, no data can be accessed. It would be great if the authors could share the complete data and materials (from SM) to enhance the reproducibility of their research.

I believe that this research represents an important research avenue that is crucial for raising public awareness about antimicrobial resistance.

I wish the authors the best of luck in their future research!

Reviewer #3 (Remarks to the Author):

This a revised version. In my reading, the manuscript has productively engaged with the reviewer feedback. I have the following comments.

The term memorability should be used consistently and not replaced by passive memory in table headings.

Memorability (dichotomous: incorrect = 0, correct = 1) and risk perception are used as both IV and DV, which is fine assuming that there is no adequate rationale for selecting either a IV or a DV role for them. The correlations (shown in Table 5 and Table 6) between the two variables in both studies (-.07 and -.106) suggests a negative relationship, that is, more accurate memory reduces perception of risk. The results of the regression analyses (Table 6 and Table 7) also show the same. Is this a correct reading? If yes, what explains this negative relationship and what are the implications?

Dear Dr Rawson,

We thank you and the three anonymous reviewers for considering the revised version of our manuscript. We are delighted to hear that you have decided to publish a suitably revised version in Communications Medicine.

We now had a chance to prepare a second revision of our paper and made final changes as per the reviewers' suggestions. Below follows our detailed response to individual reviewer comments.

Yours sincerely,
Eva Krockow

Reviewer #1:

I want to thank the authors for their thorough revisions and also their rebuttals. I believe the manuscript has improved enormously and will be a great addition to the literature and help AMR researchers to use the most appropriate terminology in their studies or when conducting science communication to broader audiences.

I am content with all changes that were made and recommend publication of the manuscript. I just have one final comment that the authors might or might not consider. With regards to my initial 10th comment about the type of regression, I am aware that in some disciplines the terminology also used in this manuscript is standard. From my experience, this stems from a misinterpretation of step-wise regression, which in these disciplines is considered only a data driven optimization method which might sometimes be understood as data tweaking. But from a pure methodological standpoint, step-wise just means computing nested regression models one after the other and comparing their delta R^2 , and the step-wise regression optimization algorithms are just one of many many ways to use step-wise regressions. Given that hierarchical clearly sometimes (in more methodological disciplines) is confused with multilevel modeling, I still recommend to change the term. However, considering different traditions between disciplines, this is optional I guess.

Response: Many thanks for these positive comments. We appreciate the additional explanation with regard to terminology around regression analyses agree we are likely talking at cross purposes due to different backgrounds.

As a compromise, we have stuck with the term hierarchical, but explicitly informed the reader that this refers to how predictors were entered into the model (p.15).

Reviewer #2:

Thank you to the authors for carefully revising the paper. They have successfully addressed most of my concerns and comments. I am happy to endorse the publication pending some minor revisions. I have only a few minor issues that I kindly ask the authors to address.

Response: Many thanks for this positive evaluation and endorsement.

Please modify the sentence in the abstract: "The necessary behaviour change relies on effective public health communication, but previous information campaigns and media coverage have failed to raise public awareness." I don't think this is a true statement given the existing evidence (see Huttner, 2010 and Thoolen, 2012 for a narrative review and meta-analysis) of the effect of public campaigns on improving knowledge and attitudes towards AMR. However, the effectiveness of the campaign could surely be improved.

Response: *This is a fair point. We have changed the sentence to: “The necessary behaviour change relies on effective public health communication, but previous information campaigns and media coverage—while showing some successes—have fallen short in generating a lasting, global increase of public awareness.”.*

I recommend avoiding the term "word effectiveness" in the abstract. The term is not self-explanatory for non-linguists and could be confusing. While it is well-defined in the paper, readers who only read the abstract may not understand it. I suggest replacing the term in the abstract with more self-explanatory terms like "memorability" and "risk association."

Response: *Thank you for pointing this out, we agree. We have replaced the term effectiveness as suggested.*

The OSF link was created to share the data, but currently, no data can be accessed. It would be great if the authors could share the complete data and materials (from SM) to enhance the reproducibility of their research.

Response: *Apologies for this. We have checked and corrected the privacy settings of the data file. The data and materials are now fully available.*

I believe that this research represents an important research avenue that is crucial for raising public awareness about antimicrobial resistance.

I wish the authors the best of luck in their future research!

Response: *Thank you!*

Reviewer #3:

This is a revised version. In my reading, the manuscript has productively engaged with the reviewer feedback. I have the following comments.

Response: *Many thanks for this positive comment.*

The term memorability should be used consistently and not replaced by passive memory in table headings.

Response: *Thank you for pointing this out, we agree. We have replaced the term “passive memory” in the two table captions.*

Memorability (dichotomous: incorrect = 0, correct = 1) and risk perception are used as both IV and DV, which is fine assuming that there is no adequate rationale for selecting either a IV or a DV role for them. The correlations (shown in Table 5 and Table 6) between the two variables in both studies (-.07 and -.106) suggests a negative relationship, that is, more accurate memory reduces perception of risk. The results of the regression analyses (Table 6 and Table 7) also show the same. Is this a correct reading? If yes, what explains this negative relationship and what are the implications?

Response: *Many thanks for your careful check of the correlation tables. Your comment prompted us to take another very close look at our analyses and results. We noticed a small coding error in our data set, which explained the counterintuitive negative relationships reported in the table; some of the dichotomous variables had been coded using the values “1” and “2” instead of “0” and “1”. This led to some relationships being shown as negative instead of positive. We corrected the coding error and re-ran all correlations. We also ensured to update our OSF page with the correctly coded data set. The correlation tables now show positive relationships between memory and risk perception. We also re-ran the regression analyses reported in the main manuscript. There were some small changes in the values previously reported and we corrected these in the tables and narrative Results section.*

However, no new significant correlation coefficients emerged and all main findings therefore remained unchanged. When correcting the values of the regression tables, we made one further change to the tables in removing the sub-headings for Blocks 1 and 2 as only the final regression Block 3 is relevant.